



# Influence of atmospheric rivers and associated weather systems on precipitation in the Arctic

Melanie Lauer[1], Annette Rinke[2], Irina Gorodetskaya[3], Michael Sprenger[4], Mario Mech[1], and Susanne Crewell[1]

[1]Institute for Geophysics and Meteorology, University of Cologne, Cologne, Germany
[2]Alfred Wegener Institute, Helmholtz Centre for Polar and Marine Research, Potsdam, Germany
[3]Interdisciplinary Centre of Marine and Environmental Research of the University of Porto, Portugal
[4]Institute for Atmospheric and Climate Science, ETH Zurich, Zurich, Switzerland

**Correspondence:** Melanie Lauer (melanie.lauer@uni-koeln.de)

**Abstract.** In this study, we analyse the contribution of Atmospheric Rivers (ARs), cyclones, and fronts to the total precipitation in the Arctic. We focus on two distinct periods of different weather conditions from two airborne campaigns: ACLOUD (May/June 2017) and AFLUX (March/April 2019). Both campaigns covered the northern North Atlantic sector, the area in the Arctic that is affected by the highest precipitation rates. Using ERA5 reanalysis, we identify pronounced regional anomalies

with enhanced precipitation rates compared to the climatology during ACLOUD due to these weather systems, whereas during AFLUX enhanced precipitation rates occur over most of the area.

We have established a new methodology, that allows us to analyse the contribution of ARs, cyclones, and fronts to precipitation rates based on ERA5 reanalysis and different detection algorithms. Here, we distinguish whether these systems occur co-located or separately. The contributions differ seasonally. During ACLOUD (early summer), the precipitation rates

are mainly associated with AR- (40%) and front-related (55%) components, especially if they are connected, while cyclone-related components (22%) play a minor role. However, during AFLUX (early spring) the precipitation is mainly associated with cyclone-related components (62%). For both seasons, snow is the dominant form of precipitation, and the small rain occurrence is almost all associated with ARs. About one-third of the precipitation can not be attributed to one of the weather systems, the so-called residual. While the residual can be found more frequently as convective than as large-scale precipitation,

the rare occasion of convective precipitation (roughly 20%) can not completely explain the residual. The fraction of precipitation classified as residual is reduced significantly when a precipitation threshold is applied that is often used to eliminate "artificial" precipitation. However, a threshold of 0.1 mm h$^{-1}$ reduces the total accumulated precipitation by a factor of two (ACLOUD) and three (AFLUX) especially affecting light precipitation over the Arctic Ocean. We also show the dependence of the results on the choice of the detection algorithm serving as a first estimate of the uncertainty.

In the future, we aim to apply the methodology to the full ERA5 record to investigate whether the differences found between the campaign periods are typical for the different seasons in which they were performed and whether any trends in precipitation associated with these weather systems can be identified.



## 1 Introduction

During the last four decades, the increase in the Arctic mean near-surface air temperature is nearly a factor of four higher than that of the global mean (Rantanen et al., 2022). This phenomenon is known as Arctic Amplification (Serreze and Barry, 2011; Wendisch et al., 2017). Evidence that the Arctic is warming includes the melting of sea ice, the retreat of glaciers, and the thawing of permafrost (Castellanos et al., 2022). The Arctic warming is forced by many processes and feedback mechanisms such as the lapse rate and snow- and ice-albedo feedback, the increasing downward longwave radiation caused by clouds

and water vapour, the reduction of sea ice in summer and the poleward heat and moisture transport (Serreze and Barry (2011); Bintanja and van der Linden (2013); Pithan and Mauritsen (2014); Sejas et al. (2014)). However, the knowledge of the involved processes and the relative importance of the feedback mechanisms is still limited.

In general, Arctic warming affects the hydrological cycle and leads to an increase in the precipitation in the Arctic (Bintanja, 2018; Boisvert and Stroeve, 2015; Vihma et al., 2016; McCrystall et al., 2021). This results in an increased total amount of

water vapor (Rinke et al., 2019), related to increased moisture holding capacity by warmer air (Bintanja, 2018), enhanced local evaporation due to the reduced sea ice cover (Bintanja and Selten, 2014), and increased poleward moisture transport from lower latitudes (Zhang et al., 2013; Gimeno et al., 2015; Bintanja et al., 2020).

Although precipitation plays a key role in the Arctic climate system, an accurate Arctic-wide observational assessment of rain and especially snowfall is still a challenge nowadays (von Lerber et al., 2022) which particularly holds for the identification

of trends (McCrystall et al., 2021). The consequence of the warming in the Arctic is not only an increase in precipitation but also a phase change from snow to rain (Bintanja and Andry, 2017; Lupikasza and Cielecka-Nowak, 2020). Therefore, rain is expected to be the dominant type of precipitation in the Arctic (Bintanja and Andry, 2017). Consequently, an increase in rain on snow and ice surfaces leads to a lower albedo that forces the snow-albedo feedback and causes sea ice melting (Perovich et al., 2002).

Poleward moisture transport is often associated with Atmospheric Rivers (ARs). ARs are long (>2000 km in length) and narrow (<1000 km in width) bands of anomalous moisture amount and transport, which can rapidly transport moisture and heat from lower latitudes to the mid-latitudes and polar regions (Ralph et al., 2020). Using the AR global detection algorithm by Guan and Waliser (2015) applied to MERRA2 reanalysis, Nash et al. (2018) found that they only cover about 10% of the Earth's surface circumference but are responsible for more than 90% of the poleward moisture transport in and across mid-

latitudes. ARs play an important role in many regions hydroclimate (Lavers and Villarini, 2015; Waliser and Guan, 2017; Viale et al., 2018). In the Arctic, ARs can bring extreme warming events both via strong heat advection and increase longwave cloud forcing (Neff et al., 2014; Komatsu et al., 2018; Mattingly et al., 2020; Bresson et al., 2022), as well as strong precipitation, including both snowfall and rainfall (Mattingly et al., 2018; Viceto et al., 2022).

Formation and existence of the ARs have been related to extra-tropical cyclones (Ralph et al., 2020; Dacre et al., 2015), warm

conveyor belts (WCBs) (Dacre et al., 2019), and tropical moisture exports (TMEs) (Bao et al., 2006; Hu and Dominguez, 2019).



While ARs, WCBs, and TMEs are interrelated, they also have distinct features: ARs can exist without WCB or TME, but they can also co-exist with TME feeding an AR with moisture, while WCB being the moisture sink due to the isentropic ascent and precipitation formation (Ralph et al., 2018). ARs can also influence the formation of extra-tropical cyclones (Sodemann and Stohl, 2013; Zhang and Ralph, 2021; Eiras-Barca et al., 2018). Guo et al. (2020) and Zhang et al. (2019) also highlighted the strong association of AR events to extratropical cyclones pointing to specific features such as the importance of anticyclone and the role of pressure gradient in the AR strength. Mo (2022) reviewed the history of the evolution of the WCB and AR concepts and their relationship to the earlier developments of a "moisture tongue" theory.

The core of the water vapour transport is concentrated in the first 2-2.5 km above ground typically in the pre-cold-frontal part of the extra-tropical cyclone (Ralph et al., 2017). At the same time, precipitation formation is often triggered by uplift along fronts and by WCBs and thus can form above the AR core. Catto et al. (2015) investigated the relation between WCBs and frontal features with respect to extreme precipitation. They found that about 70% of WCBs are linked to cold fronts during winter, and about 50% of WCBs are associated with warm fronts in the northern hemisphere. However, their study excluded the high Arctic (> 80º N) and our study aims to fill in the gap.

The purpose of our study is to determine the origin of Arctic precipitation on the synoptic scale. For this purpose, we aim to identify which precipitation is mainly associated with ARs and compare its association with cyclones and frontal zones.

We exemplary focus on two periods coinciding with airborne campaigns recently performed in the northern North Atlantic sector of the Arctic. This region encompasses the Atlantic pathway which is prone to the strongest moisture intrusions (Nash et al., 2018). The campaigns were performed at and around Svalbard within the framework of the Collaborative Research Center TR172 "Arctic Amplification: Climate Relevant Atmospheric Surface Processes, and Feedback Mechanisms (AC)[3]" (Wendisch et al., 2017). While the ACLOUD (Arctic Cloud Observations Using airborne measurements during polar Day) campaign (Wendisch et al., 2019) took place in early summer (May/June) in 2017, the AFLUX (Aircraft campaign Arctic Boundary Layer Fluxes) campaign (Mech et al., 2022) was performed in early spring (March/April) in 2019. For both campaigns, we investigate the occurring ARs in depth and develop a methodology to detect individual contributions to precipitation which can be applied to the long-term reanalysis data set in the future to investigate long-term changes in synoptical precipitation characteristics, and thus the role of air mass transport into the Arctic.

The main objective of this study is to quantify the relative contribution of ARs, cyclones, and frontal systems to precipitation in the Arctic. For this purpose, we use ERA5 data (Hersbach et al., 2020) and develop a new method to separate the precipitation within the AR shape from the precipitation related to cyclones and fronts (Sec. 2). After a comparison of the specific campaign conditions to the long-term climatology (Sec. 3.1), we quantify the precipitation associated with each of these systems and its variability for both periods (Sec. 3.2 - 3.4). Furthermore, we evaluate the precipitation types and phase partitioning of precipitation (Sec. 3.5) and assess the impact of different detection algorithms for ARs and cyclones (Sec. 3.6). The study concludes with a discussion and outlook to future work (Sec. 4).



## 2 Data and Methods

We chose two time periods with frequent AR occurrences that were encountered during the two campaigns performed at
different seasons, i.e. 28 May - 11 June 2017 (ACLOUD, 14 days) and 18 March - 6 April 2019 (AFLUX, 19 days). Both
campaigns took place around Svalbard within the Atlantic AR corridor (Nash et al., 2018) which is also associated with some
of the highest precipitation rates of the Arctic (McCrystall et al., 2021). However, as also a Siberian origin has been proposed
as a common pathway of ARs (Komatsu et al., 2018), we select the area 70°N to 90°N and 50°W to 80°E for our study (Fig. 1).
For the detection and analysis of ARs and the associated weather systems, we use reanalysis data and apply different algorithms
which are described in the following sections. Their performance is illustrated in a case study on 20 March 2019 at 00UTC,
where moist air from the south was steered northward over the North Atlantic driven by a cyclone located in North-western
Greenland (Fig. 1).

### 2.1 Reanalysis Data

All analyses in this study are based on the global reanalyses dataset ERA5 (Hersbach et al., 2020) from the European Centre of
Medium Weather Forecast (ECMWF). The data for this reanalysis is available from 1979 to the present. They have a temporal
resolution of 1 hour and a spatial resolution of 0.25° x 0.25° corresponding to ∼31 km. Specific humidity and horizontal
wind at 21 pressure levels from 1000 to 300 hPa are used to calculate the integrated water vapor transport ($IVT$) as well
as the integrated water vapor ($IWV$). Precipitation type (rain/snow) is provided as well as total, convective, and large-scale
precipitation. ERA5 gives surface precipitation as total accumulated precipitation in mm over the last hour for each grid point.
For better comparability also with other studies, this is converted to mm per day [mm day$^{-1}$] for most of the analysis. The
area-wide precipitation averages are computed as an area-weighted average.

### 2.2 Methods: Detection of atmospheric rivers and associated weather systems

For the detection of ARs, cyclones, and fronts (Sec. 2.2.1 - 2.2.3) we use existing detection algorithms. As ARs are dynamically
linked to extratropical cyclones and fronts, we make a final classification in which we define co-located and separately occurring
components (Sec. 2.2.4).

#### 2.2.1 Detection of Atmospheric Rivers

During the last years, different AR algorithms were developed. In this study, we apply the global AR detection algorithm
originally introduced by Guan and Waliser (2015) in its second version (Guan et al., 2018) (AR_Gu in the following) for our
standard setting as it is a frequently used algorithm for worldwide application. Furthermore, we test the sensitivity of the results
using a second AR algorithm developed by Gorodetskaya et al. (2014, 2020) (AR_Go in the following) specifically developed
and applied for the cold and low-moisture conditions of Antarctica and adapted for AR identification in the Arctic (Viceto
et al., 2022).



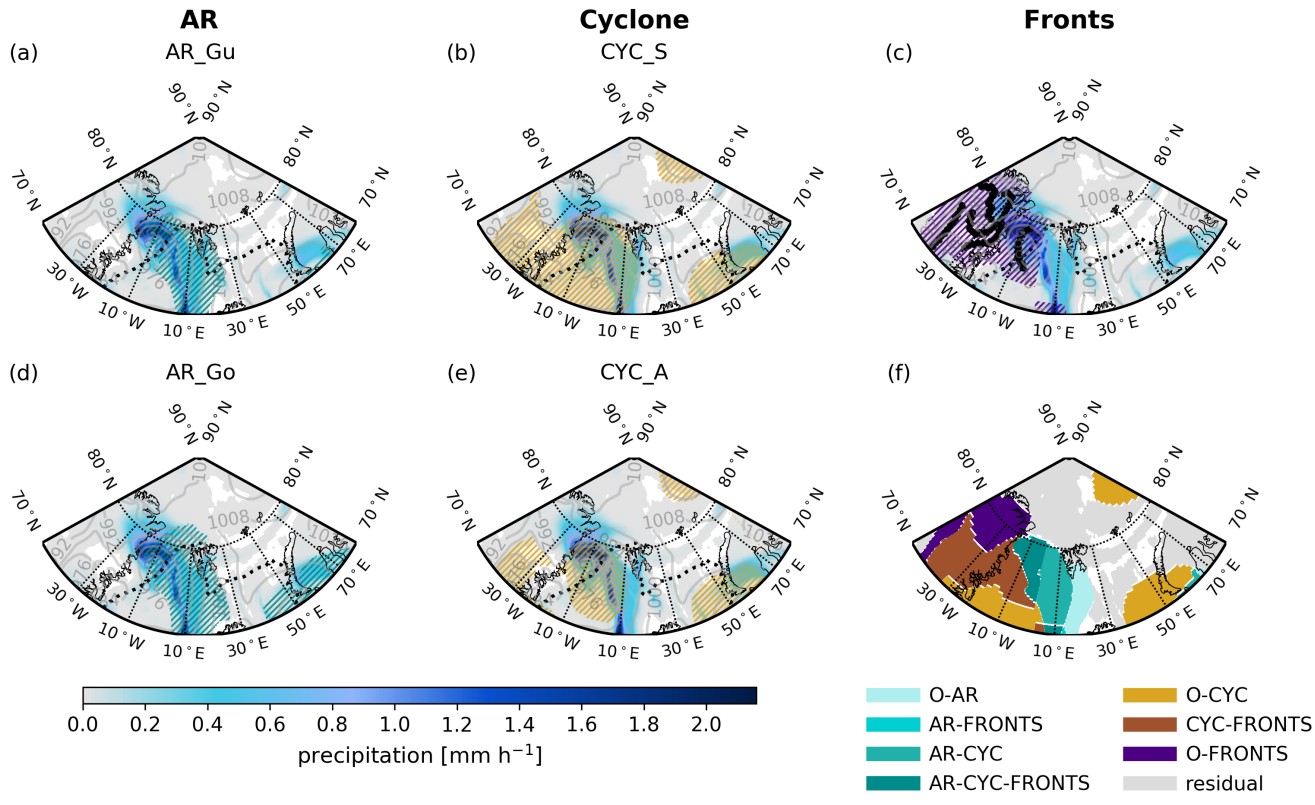

**Figure 1.** Precipitation rate in mm h$^{-1}$ on 20 March 2019 at 00UTC from ERA5 over our study area. The detected areas of ARs (cyclones) are hashed in blue (yellow). AR detection by Guan et al. (2018) (a, AR_Gu) and Gorodetskaya et al. (2014, 2020) (d, AR_Go), and cyclone detection from Sprenger et al. (2017) (b, CYC_S) and Akperov et al. (2007) (e, CYC_A). The black bold lines represent the detected fronts and the purple hashed areas represent the area of these fronts (c). The dotted black line indicates the sea ice edge based on 15% sea ice concentration, and the sea level pressure (hPa) is shown in grey isolines in (a) - (e). Classification (f) according to the GuS (AR_Gu and CYC_S) standard configuration.





The AR_Gu detection algorithm considers a combination of intensity thresholds of $IVT$ and geometry. The zonal $(x)$ and meridional $(y)$ components of the $IVT$ are calculated by using the zonal $(u)$ and meridional $(v)$ wind, the specific humidity $(q)$
profiles, the gravitational acceleration $g$ and the pressure $p$:

$$IVT_x = -\frac{1}{g} \int\limits_{1000hPa}^{300hPa} u \cdot q \cdot dp \qquad (1)$$

and

$$IVT_y = -\frac{1}{g} \int\limits_{1000hPa}^{300hPa} v \cdot q \cdot dp. \qquad (2)$$

For the $IVT$ threshold, a combination of a specific percentile and a fixed lower limit is used. In the first version of their
algorithm, Guan and Waliser (2015) first calculate the 85th percentile of $IVT$ for each grid cell from 1997-2014. The $IVT$ must exceed this percentile and the lower limit of $100 \, \mathrm{kg \, m^{-1} \, s^{-1}}$. However, due to the lower moisture capacity of the polar regions, the lower limit in these regions is set to $> 50 \, \mathrm{kg \, m^{-1} \, s^{-1}}$. These lower IVT threshold criteria make the AR_Gu algorithm too permissive in the polar regions compared to the polar-specific algorithms (Shields et al., 2022). Further requirements to detect the object as an AR are the $IVT$ direction and geometry. The $IVT$ direction has to be within 45° of the detected AR axis, the
length has to be larger than 2000 km, and the length-to-width ratio should be higher than two.

In case an object exceeds the $IVT$ percentile threshold but does not fulfill geometrical (e.g. too wide) or directional criteria, AR_Gu includes a modification of this part of the algorithm in their second version (Guan et al., 2018). This modification is similar to the concept by Wick et al. (2013). A more stringent criterium is applied if the geometrical criteria reject the detected object as an AR that has been detected via the 85th percentile. First, the 87.5th percentile threshold is used to identify the
possible AR grid cells. In case the geometrical criteria are still not met, the process is repeated for the 90th, 92.5th, and 95th percentile. In this way, it is possible to detect an AR that is surrounded by an increased moisture content.

Because we want to perform our analysis with 1 h resolution we could not make use of any existing AR catalogue. Therefore, we applied the AR_Gu algorithm to ERA5 reanalysis. To do so we calculated all relevant variables, i.e., the zonal and meridional components of the IVT ($IVT_x$ and $IVT_y$), as well as the IVT percentiles for each grid cell from 1979 - 2020.
Subsequently, we apply the algorithm to these variables and we can detect ARs for the entire ERA5 period. Figure 1 (left) illustrates the shape of an AR event detected during AFLUX on 20 March 2019 00UTC together with the surface precipitation field from ERA5.

In order to investigate the sensitivity of our results to the AR detection technique, we perform the same analyses using the AR_Go detection algorithm. This algorithm considers a combination of threshold and geometry constraints. The threshold is
based on the zonal saturated IWV ($\mathrm{IWV}_{sat}$) thus taking into account the lower saturation capacity of the polar troposphere with an AR coefficient ($\mathrm{AR}_{coef} = 0.2$) determining the relative strength of an AR (Gorodetskaya et al., 2014, 2020). For the AR analysis, we first calculated IWV and $\mathrm{IWV}_{sat}$ between pressure levels from 1000 (or nearest surface level) to 300 hPa:



$$IWV = -\frac{1}{g} \int\limits_{1000hPa}^{300hPa} q \cdot dp \tag{3}$$

and

$$IWV_{sat} = -\frac{1}{g} \int\limits_{1000hPa}^{300hPa} q_{sat} \cdot dp. \tag{4}$$

AR_Go determined an object as a potential AR when IWV is equal to or higher than this threshold:

$$IWV \geq IWV_{sat,mean} + AR_{coef} \cdot (IWV_{sat,max} - IWV_{sat,mean}) \tag{5}$$

where $IWV_{sat,mean}$ is the zonal mean IWV$_{sat}$ along each latitude, and $IWV_{sat,max}$ is the maximum value of IWV$_{sat}$ along the same latitude. Further, the object has to reach and cross 70° N and the IWV is continuous at all latitudes for at least 200 km within a maximum width of 40° longitude (Viceto et al., 2022).

### 2.2.2 Detection of Cyclones

The cyclones used in this study are derived from two different detection algorithms which apply the sea level pressure (SLP)-based method. The detection algorithm from Wernli and Schwierz (2006) and refined by Sprenger et al. (2017) (CYC_S in the following) is used in our standard configuration, while the algorithm from Akperov et al. (2015) (CYC_A in the following) is used for sensitivity testing.

For the detection of cyclones in CYC_S, a local SLP minimum is determined. If the SLP minimum is smaller than the value of the eight surrounded grid points, the grid point with the SLP minimum is considered a cyclone center. For every local SLP minimum, the outermost closed SLP contour is determined. For this purpose, the algorithm searches for every local SLP minimum enclosing contours with a pressure interval of 0.5 hPa. Further, they applied an elevation filter of 1500 m. The detected cyclones are available on a 0.5° grid, and we interpolated them to the 0.25° ERA5 grid.

CYC_A is based on Bardin and Polonsky (2005) and Akperov et al. (2007) with some modifications for the Arctic. As CYC_S, the algorithm is based on SLP and identifies the cyclone center by the minimum in SLP. To detect the outermost closed isobar they used a pressure interval of 0.1 hPa. If the pressure no longer increases, the points are defined as the outermost closed isobar. For the Arctic, the following conditions are applied: All cyclones with a size less than 200 km or a depth less than 2 hPa have been excluded. In addition, cyclones that appear or pass over regions with surface elevations higher than 1000 m are also excluded (Akperov et al., 2018).

### 2.2.3 Detection of frontal systems

The identification of fronts is based on previous studies by Jenkner et al. (2010) and Schemm et al. (2015), who mainly focused on mid-latitudes but developed a worldwide dataset. For the detection of fronts the horizontal gradient of the equivalent





potential temperature ($\nabla\theta_e$) at 700 hPa is determined and a threshold of 4 K 100 km$^{-1}$ is applied. It needs to be noted that the threshold is arbitrary and that Rüdisühli et al. (2020) considered different thresholds to account for the strong seasonal cycle of humidity. Thus a test on the sensitivity is provided below. Though a classification into warm or cold front is also provided, we do not use this information due to the frequent occurrence of occluded fronts in the Arctic.

Once the frontal line is defined the question remains which area around the frontal line should be considered to be associated
with frontal precipitation. The ascent along the surface front typically covers more than 100 km and depends strongly on the airmass. Thus, we test various distances using residual precipitation, i.e. precipitation that is not attributed to ARs, cyclones, or fronts, as a measure of the impact. Here the reasoning is that a too-short distance would provide a large residual. Accordingly, we test areas between 139 and 250 km in all directions to the front. Figure 2 illustrates how the residual precipitation declines if larger frontal areas are considered. The effect is roughly 20 % of the residual and is slightly larger in absolute terms during
ACLOUD (about 8 % of the total precipitation) compared to AFLUX (about 5 %) as residual precipitation is more frequent during ACLOUD. Based on these results we decided on a mean distance of about 200 km.

The residual is also used to study the dependence on $\nabla\theta_e$, where we test 4, 5 and 6 K 100 km$^{-1}$. Figure 2 illustrates that the residual varies less for this gradient than for the frontal distance. Therefore we stay with the original value of 4 K 100 km$^{-1}$.

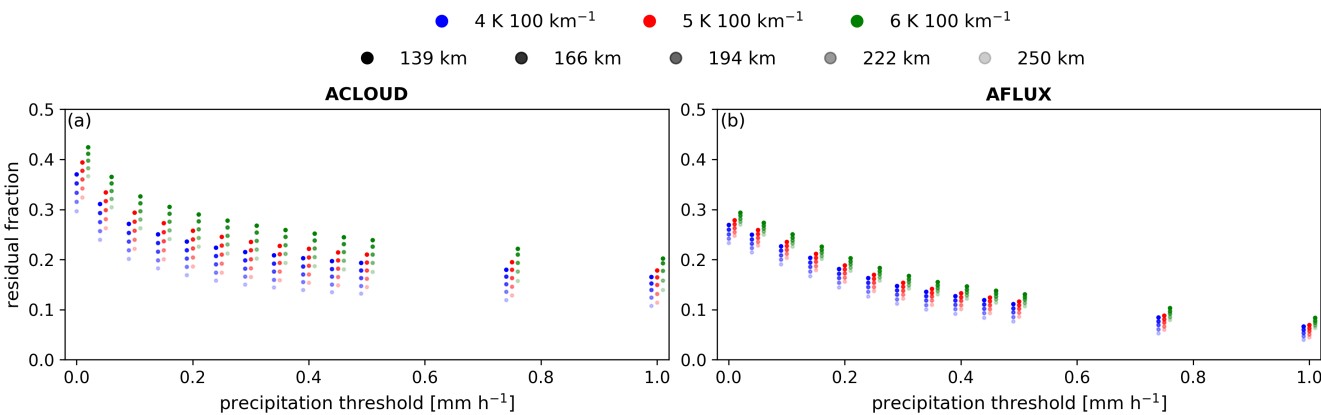

**Figure 2.** Fraction of residual precipitation for ACLOUD (a) and AFLUX (b) for different thresholds of precipitation [mm h$^{-1}$] related to the gradient of the equivalent potential temperature (blue: 4 K 100 km$^{-1}$, red: 5 K 100 km$^{-1}$, green: 6 K 100 km$^{-1}$), and the distance (139, 166, 194, 222 and 250 km) from the detected front line (distance increases from dark to light shaded dots). In the calculation of the residual the standard configuration, GuS (AR_Gu and CYC_S) was used.

### 2.2.4 Final Classification

For each ERA5 time step the techniques to detect ARs (Sec. 2.2.1), cyclones (Sec. 2.2.2), and frontal areas (Sec. 2.2.3) are applied to assign the appropriate system to each grid cell. Figure 1 shows exemplarily the results of the different detection schemes for 20 March 2019 at 00UTC. An AR produces precipitation along the Atlantic pathway and is detected by both AR





algorithms with some differences in the position such as a slightly more northern extent produced by AR_Go than in AR_Gu. The AR is steered by an intense cyclone in the west which covers a larger region in CYC_S than in CYC_A. Connected to that 195 are frontal regions over northeast Greenland.

We classify each grid cell using our standard configuration AR_Gu and CYC_S (GuS in the following). From Fig. 1, it becomes obvious that certain grid cells can be assigned not only to one but to multiple weather systems. Only in the region 5 - 25° E south of Svalbard, the AR is solely responsible (O-AR) for precipitation while in the western part, it is connected with a cyclone (AR-CYC), frontal areas (AR-FRONTS), and all systems together (AR-CYC-FRONTS). Considering all weather 200 systems, we can identify in total seven different components which further include only cyclones (O-CYC), cyclones co-located with fronts (CYC-FRONTS), and only fronts (O-FRONTS).

Due to the fact that ARs, cyclones, and fronts are dynamic features associated with strong winds, precipitation may fall outside the identified shapes. Furthermore, we can see that the shapes of the detected weather systems can differ among the algorithms (Fig. 1). Precipitation outside of the area of the detected weather systems is classified as residual. The residual 205 depends on the precipitation threshold: A drastic decrease of the residual with increasing precipitation threshold can be noted (Fig. 2). Because atmospheric models are known to produce very light precipitation, often thresholds of up to $0.1 \text{ mm h}^{-1}$ are considered (Boisvert et al., 2018; Yang et al., 2021). The fact that the residual is roughly reduced by 30% during ACLOUD, and 16% during AFLUX by applying a threshold of $0.1 \text{ mm h}^{-1}$, highlights the importance of light precipitation for the Arctic. Therefore, we decided not to apply a threshold for our analyses. However, we discuss in Sec. 3.4 the sensitivity of the results 210 by applying different precipitation thresholds.

## 3 Results

The influence of different weather systems (ARs, cyclones, and fronts) on the precipitation in the Arctic is analysed in this section for the early summer (ACLOUD) and early spring (AFLUX) campaigns. First (Sec. 3.1), we investigate how precipitation during both campaigns relates to long-term climatology. Second, we investigate the contribution of ARs, cyclones, and fronts 215 to the total precipitation. For this purpose, we analyse their spatiotemporal evolution and the contribution of these weather systems (Sec. 3.2). Furthermore, we address the role of precipitation intensity (Sec. 3.3), the sensitivity of threshold (Sec. 3.4), the issue of precipitation phase (Sec. 3.5), and assess the impact of the choice of detection algorithms (Sec. 3.6).

### 3.1 Precipitation during the campaigns compared to climatology

How intense was the precipitation during the campaigns compared to the climatological perspective? To answer this question, 220 we calculate the daily averaged precipitation rate for both campaigns (Fig. 3) as well as for the climatology (1979 - 2021) over the respective period. The climatology for both periods shows a strong north-south gradient with the lowest values in the central Arctic (and also Greenland) (not shown). Low precipitation values in the central Arctic (north of 80° N) are in agreement with (McCrystall et al., 2021) who find between 0.2 and $0.8 \text{ mm day}^{-1}$ for the annual average based on ERA5.

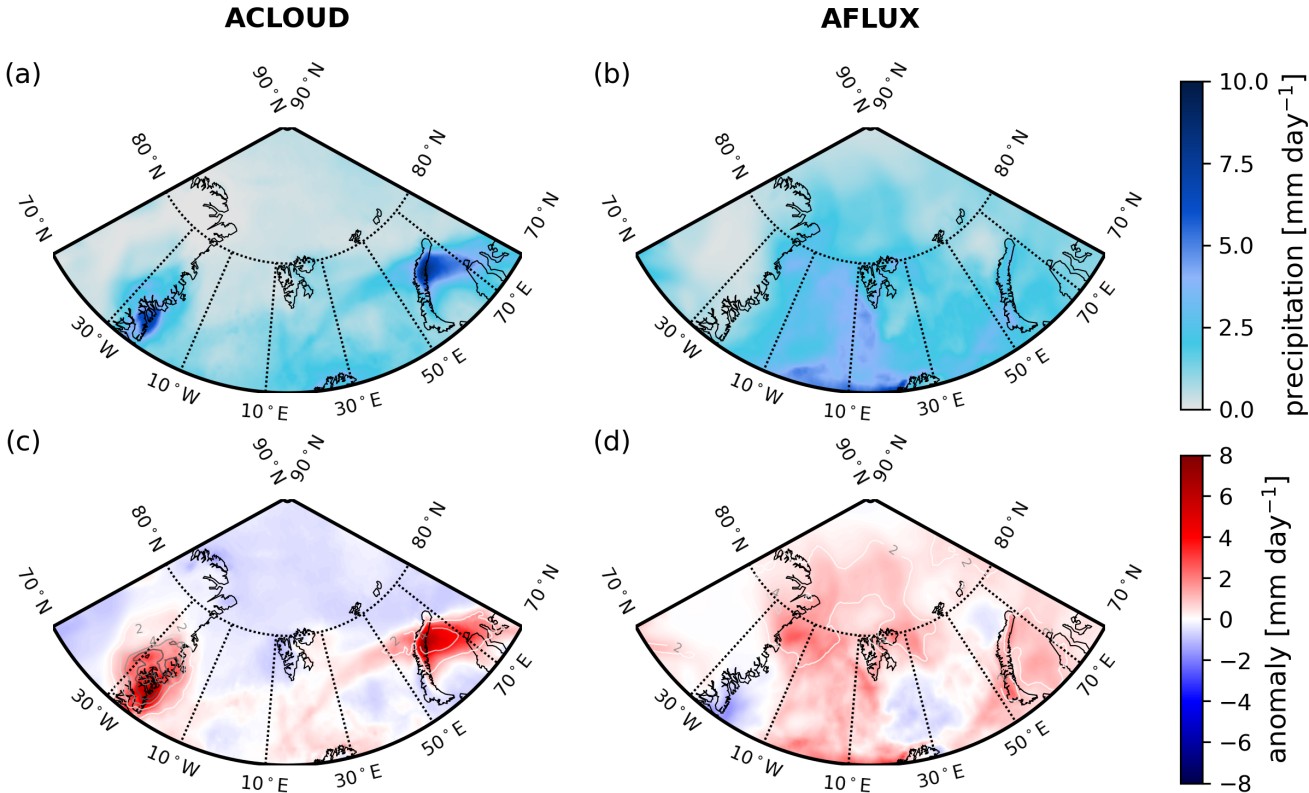

**Figure 3.** Daily averaged precipitation rate [mm day$^{-1}$] for ACLOUD (left) and AFLUX (right). For the campaign year ((a) and (b)), the anomaly with respect to the climatology, and the deviation from the climatology as contour lines ((c) and (d)).

During ACLOUD (AFLUX), the amount of precipitation in the studied area is 12% (39%) higher compared to the clima-
tological mean. For both campaigns, we can identify hot spots with enhanced precipitation likely originating from weather
systems. During ACLOUD, two clearly defined regions one on the east coast of Greenland and the other in the Kara Sea and
the northern part of Novaya Zemlya show precipitation rates of more than 4 mm day$^{-1}$, which corresponds to anomalies of
8 mm day$^{-1}$ (Greenland) and 5 mm day$^{-1}$ (Kara Sea) with respect to the climatological value. The maximum over Novaya
Zemlya extends, albeit with lower values, towards the western Atlantic indicating its origin from Siberian ARs (Viceto et al.,
2022). During AFLUX, the Atlantic corridor sticks out with the highest values and a clear positive anomaly in the Fram Strait.
In contrast to ACLOUD, the enhanced precipitation rates are distributed over most of the area, also over the central Arctic (>
80° N). The most enhanced precipitation compared to climatology (by a factor of three), however, is identified in northeast
Greenland. The regionally distinct maxima in precipitation already indicate that transient synoptical features might determine
the precipitation distribution on the time scale of weeks.



## 3.2 Contribution of ARs, cyclones, and fronts to the total precipitation

How much do ARs, cyclones, and fronts contribute to the precipitation during the two campaigns in the Arctic, especially to the hot spots shown in Fig.3? To answer this question, we use the methodology from Sec. (2.2.4) with the standard configuration GuS (AR_Gu and CYC_S) and analyse these contributions concerning their temporal variation (Fig. 4), spatial patterns (Fig. A1) and latitudinal dependency (Fig. A2).

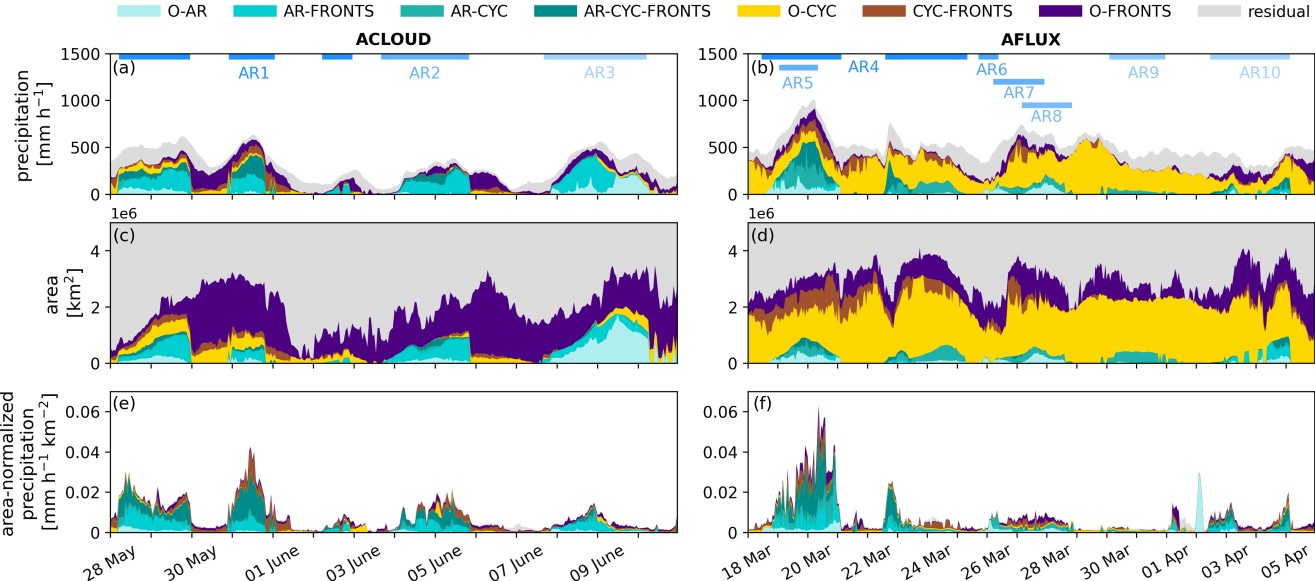

**Figure 4.** Time series of domain-accumulated hourly precipitation rate [mm h$^{-1}$] (a,b), the size of the area [km$^2$] (c,d), and the ratio between the precipitation rate and the area [mm h$^{-1}$ km$^{-2}$] (e,f) for different weather systems for ACLOUD (left, 28 May - 11 June 2017) and AFLUX (right, 18 March - 6 April 2019). The colors represent the co-located and separated components.

During ACLOUD, the daily averaged precipitation rate accumulated over the whole study domain amounts to 7.6 x 10$^3$ mm day$^{-1}$ (Table 1). Most of the precipitation was located between 70° and 80° N (Fig. A2). Considering the whole period, 40% (co-located: 31%, only: 9%) of the total precipitation can be explained by ARs, while only 22% (co-located: 14%, only: 8%) is related to cyclones. However, with 55% (co-located: 40%, only: 15%) the majority of precipitation is associated with frontal signatures in both cases, i.e. when co-located with ARs and cyclones and also if regarded if occurring alone.

In total, we identify three ARs, two originated from Siberia (AR1 and AR2), and one from the Atlantic (AR3) (Table 2). Note, that AR_Gu does have a gap in the detection of AR1 and AR2 (Fig. 4, Tab. 2). During this detection gap of AR1 and AR2, the precipitation is then associated with fronts only. The temporal development of these ARs is visualized as movies in the supplement (see video supplement Lauer (2022)). All of them reached at least 77° N (Table 2) and have been described in detail by Viceto et al. (2022). These ARs, especially in connection with other weather systems contributed to the enhanced





precipitation rates over the Kara Sea and northern Novaya Zemlya (AR1) and at the east coast of Greenland (AR3), when they made landfall (Fig. A1). The slightly higher precipitation rates over the southern part of the Barents and Norwegian Seas are caused by AR2. AR1 and AR2 were mainly co-located with fronts and cyclones (AR-FRONTS, AR-CYC-FRONTS), whereas AR3 was co-located with fronts (AR-FRONTS) or occurred alone (O-AR). Thus, most precipitation is generated when ARs
are collocated with cyclones and/or fronts (Table 1, Fig. 4).

**Table 1.** Daily averaged precipitation rate over the study domain [mm day$^{-1}$] and total precipitation area (km$^2$) as average over the ACLOUD and AFLUX periods. The contribution of ARs, cyclones, fronts as well as of the residual is given in percent for all classes involving the respective feature ("total" as $t$), for their co-location ("co-located" as $c - l$) and for their individual occurrence ("only" as $o$). Numbers are given for the different combinations of AR detection algorithms by Guan et al. (2018) (AR_Gu) and Gorodetskaya et al. (2014, 2020) (AR_Go), and the cyclone detection from Sprenger et al. (2017) (CYC_S) and Akperov et al. (2007) (CYC_A): GuS (AR_Gu & CYC_S), GuA (AR_Gu & CYC_A), GoS (AR_Gu & CYC_S), and GoA (AR_Go & CYC_A)

| | ACLOUD | | | | AFLUX | | | |
|---|---|---|---|---|---|---|---|---|
| daily precipitation (x10$^3$) [mm day$^{-1}$] | 7.6 | | | | 12.5 | | | |
| | ARs | cyclones | fronts | residual | ARs | cyclones | fronts | residual |
| | t / c-l / o | t / c-l / o | t / c-l / o | | t / c-l / o | t / c-l / o | t / c-l / o | |
| GuS | 40 / 31 / 9 | 22 / 14 / 8 | 55 / 40 / 15 | 29 | 16 / 12 / 4 | 62 / 15 / 47 | 19 / 14 / 5 | 25 |
| GuA | 40 / 32 / 8 | 28 / 18 / 10 | 55 / 42 / 13 | 28 | 16 / 8 / 8 | 41 / 8 / 33 | 19 / 12 / 7 | 38 |
| GoS | 19 / 16 / 3 | 22 / 14 / 8 | 55 / 27 / 28 | 35 | 40 / 32 / 8 | 62 / 33 / 29 | 19 / 15 / 4 | 22 |
| GoA | 19 / 17 / 2 | 28 / 17 / 11 | 55 / 30 / 25 | 33 | 40 / 22 / 18 | 41 / 21 / 20 | 19 / 13 / 6 | 30 |
| area (x10$^7$) [km$^2$] | 189 | | | | 257 | | | |
| | ARs | cyclones | fronts | residual | ARs | cyclones | fronts | residual |
| | t / c-l / o | t / c-l / o | t / c-l / o | | t / c-l / o | t / c-l / o | t / c-l / o | |
| GuS | 8 / 4 / 4 | 6 / 2 / 4 | 28 / 13 / 15 | 65 | 5 / 3 / 2 | 38 / 5 / 33 | 18 / 11 / 7 | 47 |
| GuA | 8 / 4 / 4 | 10 / 3 / 7 | 28 / 13 / 15 | 62 | 5 / 2 / 3 | 25 / 2 / 23 | 18 / 10 / 8 | 57 |
| GoS | 4 / 3 / 1 | 6 / 2 / 4 | 28 / 11 / 17 | 68 | 13 / 7 / 6 | 38 / 11 / 27 | 18 / 11 / 7 | 45 |
| GoA | 4 / 3 / 1 | 10 / 2 / 8 | 28 / 12 / 16 | 65 | 13 / 6 / 7 | 25 / 6 / 19 | 18 / 10 / 8 | 53 |

The picture changes strongly for AFLUX. During this period, the daily averaged precipitation rate of 12.5 x 10$^3$ mm day$^{-1}$ accumulated over the study domain is more than 60 % higher than during ACLOUD. The main source of precipitation is also different compared to ACLOUD (with dominating fronts). For AFLUX precipitation is mainly associated with cyclones (62%; co-located: 15%, only: 47%), especially if they occur separately (O-CYC) whereas the contribution of fronts (19%; co-located:
14%, only: 5%) and ARs (16%; co-located: 12%, only: 4%) is comparably small (Fig. 4, Table 1, Fig. A2). Thus, although seven ARs were detected during AFLUX, their contribution to the total precipitation rate is a factor of 2.5 lower compared to ACLOUD (Table 1).



**Table 2.** Detected Atmospheric River (AR) events during the ACLOUD (May/June 2017) and AFLUX (March/April 2019) campaigns. For each event, the start, end, pathway, northernmost point, and affected areas are specified.

| # of AR | start (date / time [UTC]) | end (date / time [UTC]) | pathway | furthest point (°N) | affected areas |
|---|---|---|---|---|---|
| AR1 | 28 May / 05 | 29 May / 23 | Siberian | 83.00 | Kara Sea, Barents Sea |
|  | 31 May / 00 | 31 May / 19 |  |  | Norwegian Sea |
|  | 02 June / 06 | 02 June / 21 |  |  | Norwegian Sea |
| AR2 | 03 June / 18 | 05 June / 20 | Siberian | 77.00 | Kara Sea, Barents Sea |
|  |  |  |  |  | Norwegian Sea |
| AR3 | 07 June / 16 | 10 June / 06 | Atlantic | 85.25 | Greenland, Norwegian Sea |
| AR4 | 18 March / 12 | 21 March / 02 | Labrador Sea | 86.75 | Greenland, Norwegian Sea |
|  | 22 March / 15 | 25 March / 08 |  | 77.75 | Barents Sea, Kara Sea |
| AR5 | 19 March / 01 | 20 March / 08 | Europe | 72.00 | Kara Sea |
| AR6 | 25 March / 17 | 26 March / 09 | Atlantic | 77.00 | Greenland, Norwegian Sea |
| AR7 | 26 March / 05 | 27 March / 22 | Africa | 77.75 | Kara Sea |
| AR8 | 27 March / 04 | 28 March / 20 | Labrador Sea | 76.50 | Greenland, Norwegian Sea |
|  |  |  |  |  | Barents Sea |
| AR9 | 30 March / 02 | 31 March / 23 | Siberia | 84.75 | Kara Sea |
| AR10 | 02 April / 15 | 05 April / 03 | Greenland | 81.50 | Greenland, Norwegian Sea |
|  |  |  |  |  | Barents Sea |

In contrast to ACLOUD, the ARs came mainly from the Atlantic or Labrador Sea (Table 2) and were first meridionally orientated with a subsequent zonal alignment over the studied area (see supplement Lauer (2022)). Although the majority of the ARs (5 out of 7) reached 77° N or higher, their contribution did not exceed 16%. These ARs precipitated mostly out in the lower latitudes between 60° and 67° N (Fig. A2). At higher latitudes, the AR-related precipitation rates were mainly associated with AR4 that reached up to 87° N (Table 2). This AR4 was mainly associated with cyclones (AR-CYC) and contributed to the enhanced precipitation over the Norwegian Sea and Fram Strait (Fig. 3, Fig. A1).

In summary, we hypothesize that the differences between both campaigns might be related to seasonal effects, i.e. early summer vs. early spring. First, the precipitation during ACLOUD was mainly associated with ARs and fronts, whereas during AFLUX, the precipitation was mainly concentrated within cyclones. Second, during ACLOUD, the systems were more effective when occurring in concert, i.e. the contribution of ARs and fronts is about three times higher if they occur together or with cyclones, compared to when occurring alone (Table 1). While this is also valid for ARs during AFLUX, it is not the case for cyclones whose contribution is highest when occurring alone.





## 3.3 Area and time-dependent precipitation intensity

The seasonal differences in the contribution of ARs, cyclones, and fronts can be explained, among others, by the area of the individual systems. In percentage terms, ARs and fronts cover a greater area during ACLOUD than during AFLUX, while cyclones cover a smaller area during ACLOUD than during AFLUX. This applies to both co-located and separate components (Table 1). However, a larger area does not necessarily mean higher precipitation rates (Fig. 4, Table 1, Fig. A2 ). During ACLOUD, the area covered by fronts is a factor of 3.5 (co-located: 3.3, only: 3.8) higher than the area covered by ARs. However, the precipitation rate associated with fronts is only a factor of 1.4 (co-located: 1.3, only: 1.6) higher than that associated with ARs. The same behavior is also seen during AFLUX (Table 1). In general, higher precipitation rates related to ARs are concentrated within a smaller area - independent of whether co-located or separated. Therefore, the precipitation rate with respect to the area is dominated by ARs, especially in conjunction with fronts and cyclones, during both campaigns (Fig. A2). This is surprising for fronts (during ACLOUD) and cyclones (during AFLUX) which affect a greater area than ARs (co-located and separated). Consequently, we demonstrate that the front and cyclone-related precipitation rates, during ACLOUD and AFLUX, respectively, are not as intense as AR-related precipitation rates.

To further investigate precipitation intensity, we look at how the average precipitation is distributed over the different hourly precipitation rates. Note, that for these distributions we do not weigh precipitation rates by the area of the respective grid point (Fig. 5). By treating each ERA5 grid point equally more emphasis is put on the central Arctic. The snowfall which is the dominating precipitation type during both campaigns (see Sec. 3.5) can be classified as light ($< 1$ mm h$^{-1}$), moderate ($1 - 2.5$ mm h$^{-1}$) and heavy ($> 2.5$ mm h$^{-1}$) precipitation (DWD, 2023). The light precipitation rates are mainly associated with components that are not co-located (O-AR, O-FRONTS, O-CYC). In contrast, the occurrence of the highest precipitation rates in the Arctic is most likely when different weather systems occur in conjunction. This mainly concerns precipitation associated with ARs. Especially in connection with fronts (ACLOUD) and cyclones (AFLUX), we observe moderate precipitation ($1 - 2.5$ mm h$^{-1}$) amounts. More than 92% of the AR-related precipitation can be classified as moderate precipitation during ACLOUD and AFLUX. Only a small amount of light precipitation ($< 6\%$) is related to AR-related components. Therefore, it could be possible that parts of light precipitation related to residual (or also to the other weather systems) might be in the vicinity of the detected AR shape. As the detection of ARs depends on the moisture content, the moisture might be too low to be detected as an AR. For example, the gap in the detection of AR1 on 30 May during ACLOUD (Fig. 4) might hint at this phenomenon.

There are seasonal differences regarding AR-related intensity. We find that the precipitation rates associated with ARs are more intense during ACLOUD compared to AFLUX. This could be a seasonal aspect, i.e. ARs are more intense in early summer than in early spring. However, also the orientation of the ARs could be an aspect. The ARs (AR1 and AR2) during ACLOUD originated in Siberia and were mostly zonally orientated, whereas AR3 and the ARs during AFLUX were meridionally orientated. Fig. A2 shows that these meridionally orientated ARs already lose high amounts of precipitation on their way to the Arctic.





**Figure 5.** Contribution of different precipitation rates [mm h$^{-1}$] to the daily averaged precipitation [mm day$^{-1}$] for co-located and separated components (a-h) during ACLOUD (top) and AFLUX (bottom). The accumulated daily precipitation rate [mm day$^{-1}$] is shown by the dotted line and their value is given by the y-axis. Note, that each 0.25 x 0.25 grid cell is not area-weighted. Thus, the sum of the cumulative precipitation deviates from Table 1 which includes the area-weighted precipitation. The contribution of the area-weighted precipitation is shown in Fig. A3.



## 3.4 Residual Contribution and Threshold of Precipitation

During ACLOUD and AFLUX, 29 and 25%, respectively, of the precipitation cannot be associated with any of the weather
systems (Tab. 1). Especially in the central Arctic (> 80° N), where the weather systems are quite rare or are difficult to detect,
the residual explains up to 100% of the precipitation for individual grid cells (Fig. A1). However, the occurrence of the residual
decreases with higher latitudes (Fig. A2). Fig. 5 shows that more than 95% (85%) of the residual precipitation has rates lower
than 0.1 mm h$^{-1}$ during ACLOUD (AFLUX).

Imura and Michibata (2022) have shown that dynamic models produce too light and too frequent precipitation, especially in
the Arctic. Therefore, in several studies a precipitation threshold of 0.1 mm h$^{-1}$ is used to suppress this 'artificial precipitation'
(Boisvert et al., 2018). However, others argue that light precipitation rates, especially drizzle over the Arctic Ocean would be
underestimated by using this threshold (Barrett et al., 2020). For our study, the introduction of such a threshold would not
only affect the residual but also suppress the lower precipitation rates for other categories such as AR-FRONTS, O-CYC,
CYC-FRONTS, and O-FRONTS during ACLOUD, as well as AR-CYC, CYC-FRONTS, and O-FRONTS during AFLUX.
Especially during AFLUX, this light precipitation is important as can be seen in the case of cyclones. Here, cyclones have
mostly precipitation rates below 0.1 mm h$^{-1}$ but contribute most (47%) to the total precipitation (Fig. 5).

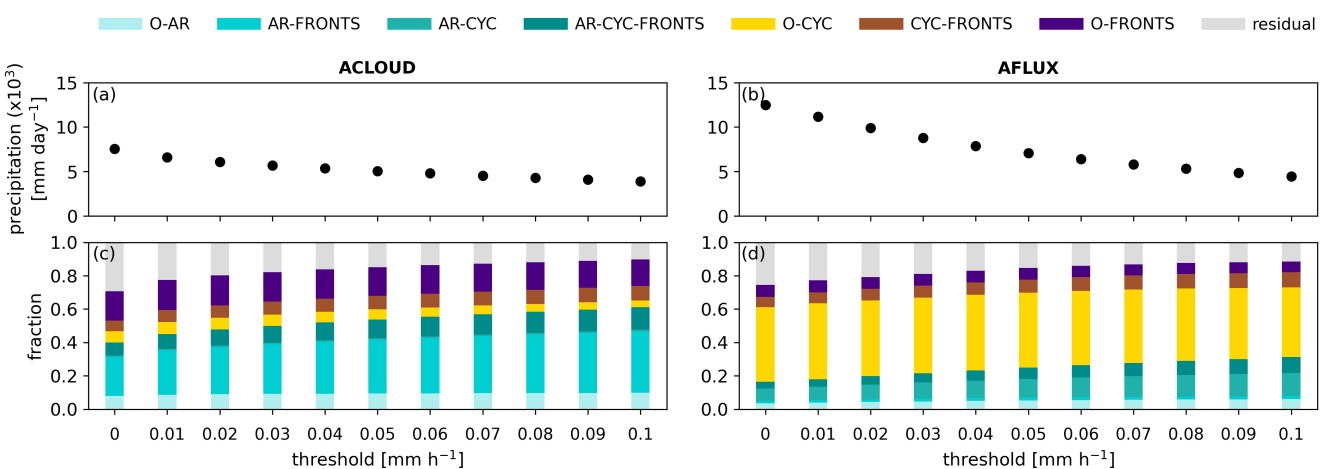

**Figure 6.** Daily averaged precipitation ((a) and (b)) and the fraction of ARs, cyclones, and fronts (co-located and separated) ((c) and (d)) for
different precipitation thresholds [mm h$^{-1}$].

Section 2.2 has already shown the sensitivity of the residual to the introduction of a precipitation threshold. Figure 2 has
shown that a threshold of 0.1 mm h$^{-1}$ instead of 0 mm h$^{-1}$ would roughly lead to a reduction of the residual fraction by
one-third. But still, the question remains how the weather systems are affected by the threshold in particular for lightest
precipitation. Figure 6 shows how drastically precipitation is reduced when the threshold is increased stepwise to 0.1 mm h$^{-1}$.
Daily precipitation decreases from 7.6 x 10$^3$ to 3.9 x 10$^3$ mm day$^{-1}$ during ACLOUD, and from 12.4 x 10$^3$ to 4.5 x 10$^3$
mm day$^{-1}$ during AFLUX. Consequently, the precipitation rate decreases by 50% during ACLOUD and 64% during AFLUX.





Furthermore, the contribution of AR-related components (ACLOUD: +21%; AFLUX: + 15%), cyclone-related components (ACLOUD: +5%; AFLUX: +12%), and front-related components (ACLOUD: +20%; AFLUX: +8%) increases, whereas the
residual decreases (ACLOUD: - 19%; AFLUX: -14%). In summary, the contribution of ARs connected with fronts (ACLOUD) and cyclones (AFLUX) would become much more dominating if a threshold would be introduced.

## 3.5 Type and form of precipitation

We now analyse the phase composition of total precipitation and its distribution for the different weather systems (Table 3). At the same time, we also investigate whether any differences with respect to convective and large-scale rain and snow exist. Snow
is the dominant type of precipitation for both campaigns with 67% for ACLOUD and 90% for AFLUX which took place in March/April exhibiting colder temperatures than ACLOUD. Considering the snowfall rates of the different weather systems we again see the stronger effect of ARs for ACLOUD (co-located: 26%, only: 7%) compared to AFLUX (co-located: 11%, only: 3%) and the clear dominance of cyclones for AFLUX (co-located: 14%, only: 58%) compared to ACLOUD (co-located: 12%, only: 9%). Nevertheless, ARs are even more important for rain than for snowfall with even higher percentages, i.e. AR fraction
amounts to 54% (co-located: 42%, only: 12%) for ACLOUD, and 42% (co-located: 29%, only 13%) for AFLUX. However, the result needs to be interpreted carefully as also the AR_Gu detection algorithm is too permissive in the polar regions which are more prone to snowfall. One indication of this hypothesis is the lower residual for rain (ACLOUD: 20%, AFLUX: 12%) than for snow (ACLOUD: 34%, AFLUX: 27%).

**Table 3.** Daily averaged, area accumulated total (tot), convective (con), and large-scale (l-s) rain and snowfall rate, as well as total precipitation (tot pres) expressed in $10^3$ mm day$^{-1}$ during ACLOUD and AFLUX. The contribution of ARs, cyclones, fronts as well as of the residual is given in percent for all classes involving the respective feature ("total" as $t$), for their co-location ("co-located" as $c - l$) and for their individual occurrence ("only" as $o$).

| | ACLOUD | | | | | AFLUX | | | | |
| | total | ARs | cyclones | fronts | residual | total | ARs | cyclones | fronts | residual |
| | rate | t / c-l / o | t / c-l / o | t / c-l / o | | rate | t / c-l / o | t / c-l / o | t / c-l / o | |
|---|---|---|---|---|---|---|---|---|---|---|
| con snow | 0.6 | 9 / 4 / 5 | 17 / 4 / 13 | 17 / 10 / 7 | 66 | 2.2 | 4 / 3 / 1 | 62 / 3 / 59 | 6 / 5 / 1 | 34 |
| l-s snow | 4.5 | 37 / 29 / 8 | 20 / 13 / 7 | 57 / 40 / 17 | 29 | 9.0 | 16 / 12 / 4 | 62 / 17 / 45 | 21 / 15 / 6 | 25 |
| tot snow | 5.1 | 33 / 26 / 7 | 19 / 12 / 7 | 52 / 36 / 16 | 34 | 11.0 | 14 / 11 / 3 | 62 / 14 / 48 | 18 / 13 / 5 | 27 |
| con rain | 0.8 | 33 / 22 / 11 | 25 / 10 / 15 | 42 / 30 / 12 | 33 | 0.7 | 21 / 12 / 9 | 72 / 14 / 58 | 9 / 7 / 2 | 17 |
| l-s rain | 1.7 | 63 / 51 / 12 | 26 / 21 / 5 | 70 / 57 / 13 | 14 | 0.6 | 67 / 49 / 18 | 73 / 50 / 23 | 31 / 30 / 1 | 5 |
| tot rain | 2.5 | 54 / 42 / 12 | 26 / 18 / 8 | 61 / 49 / 12 | 20 | 1.3 | 42 / 29 / 13 | 72 / 30 / 42 | 20 / 19 / 1 | 12 |
| tot precip | 7.6 | 40 / 31 / 9 | 22 / 14 / 8 | 55 / 40 / 15 | 29 | 12.5 | 16 / 12 / 4 | 62 / 15 /47 | 19 / 14 / 5 | 25 |

Looking at the precipitation formation mechanism it becomes clear that large-scale precipitation prevails for both campaigns
(Tab. 3) which is not surprising as the focus of this study is on dynamical weather systems. Furthermore, the campaign periods were selected for AR occurrence. For snowfall, the large-scale component dominates compared to the convective component,



which contributes less than 3% and 25% for ACLOUD and AFLUX, respectively. For rain, the fraction of total precipitation is highest for ACLOUD with 33% and lower for AFLUX with 10%. Regarding the contribution of ARs, cyclones, and fronts, we can see that ARs and fronts mainly contribute to the large-scale than to the convective rain and snow, whereas the contribution
of cyclones to the convective and large-scale rain and snow is quite similar. The residual is higher for the convective than the large-scale component, which is reasonable as large-scale precipitation should be explained by synoptic features. Thus, large-scale residuals might be connected to problems in the detection algorithms or related to dynamical features producing precipitation by vertical displacement outside the object shape.

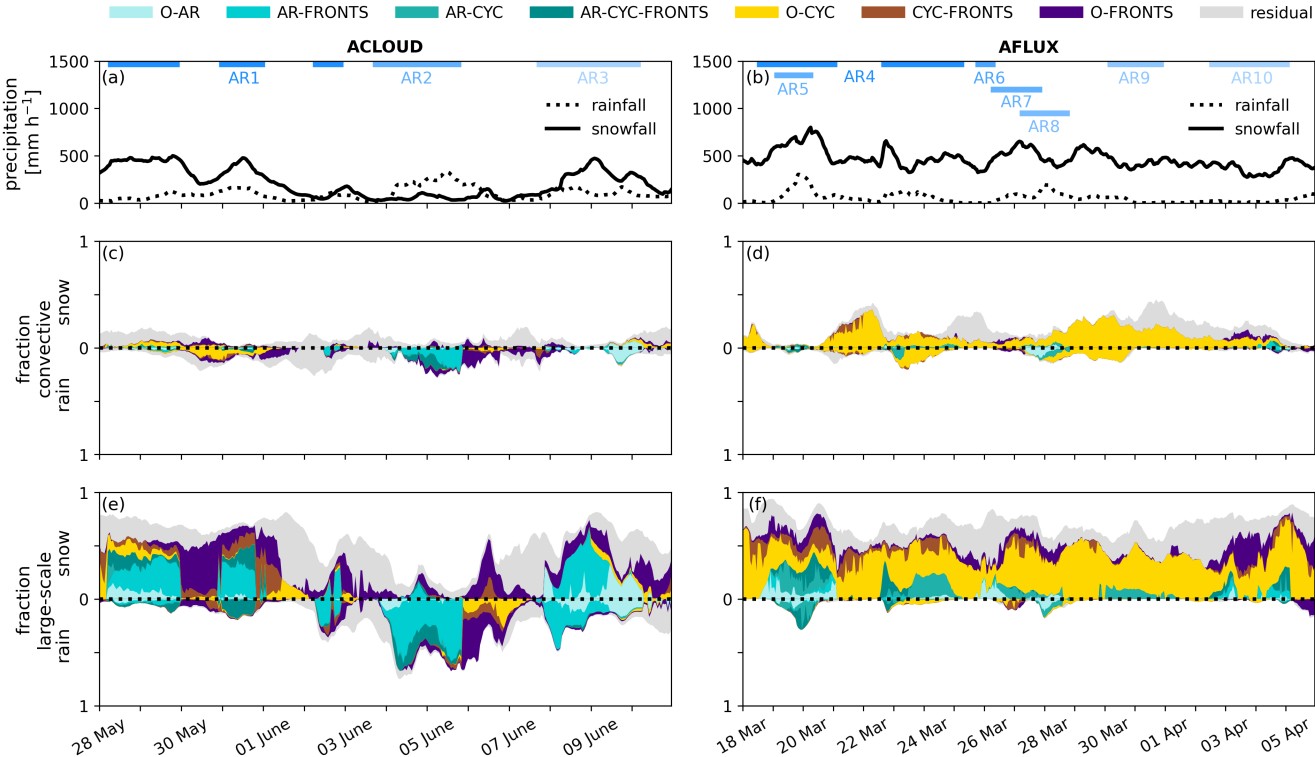

**Figure 7.** Time series of hourly rain (dotted) and snowfall (solid) [mm h$^{-1}$] ((a) and (b)) and the fraction of convective ((c) and (d)) and large-scale ((e) and (f)) rain and snow to the total precipitation for ACLOUD (left) and AFLUX (right) and for co-located and separated components (colors as in Fig. 1)

The temporal development (Fig. 7) shows that during ACLOUD a brief phase during 3-6 June occurs when rain becomes
the dominant type of precipitation although snowfall is the dominant precipitation overall. In this period, the rainfall is mainly associated with AR2 with some convective contribution. As described in the previous section, this AR moved over the southern part of the Barents and Norwegian Seas. Thus, the rainfall was mainly concentrated in the lower latitudes (70 - 75° N) (Fig. 8). On 6 June, no AR was detected and the rainfall was more related to O-FRONTS and O-CYC. During this day, there was a slight increase in snowfall. In general, the major precipitation events during ACLOUD occur when ARs were connected with



fronts (Fig. 7). In between the AR events occurrences, precipitation is low but prominently associated with residual snowfall, which might be due to weak/fading synoptical systems not detected by the algorithms.

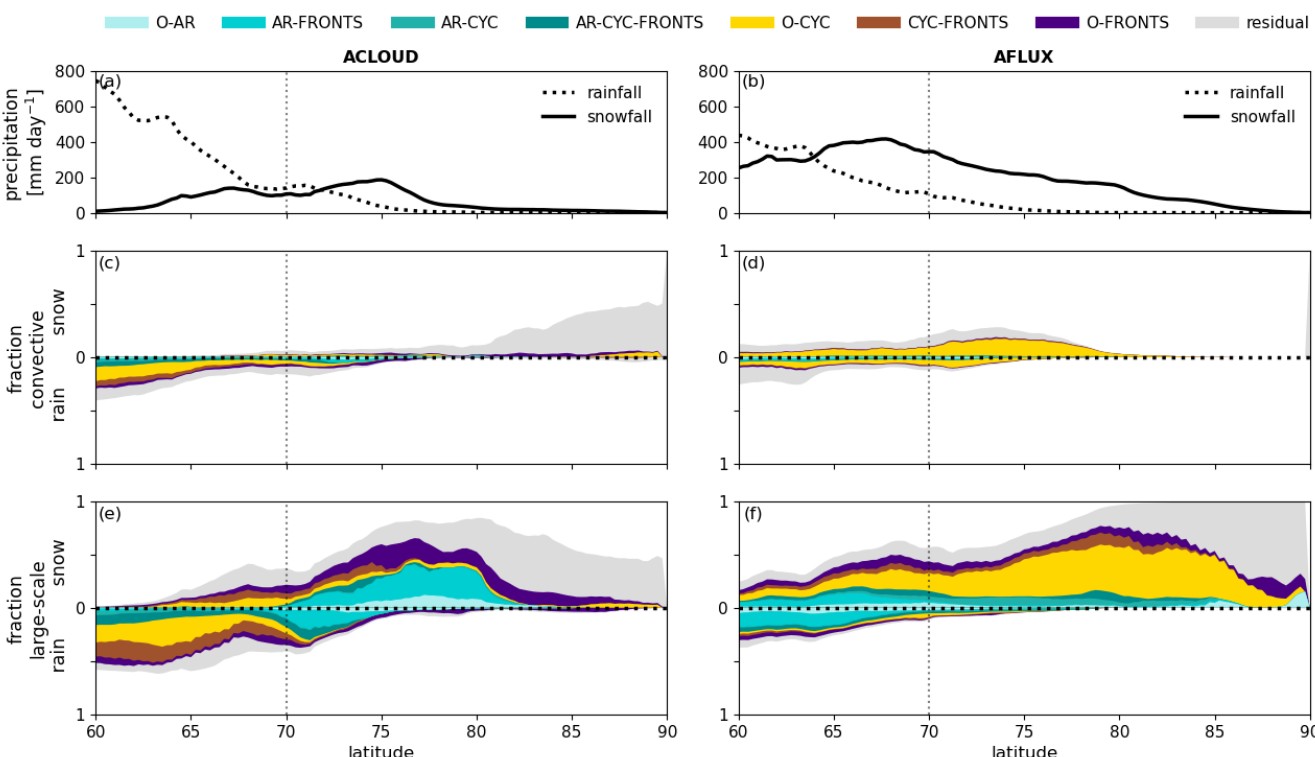

**Figure 8.** Latitudinal dependence (60 - 90° N) of daily averaged rain (dotted) and snowfall (solid) [mm day$^{-1}$] (a,b) for ACLOUD (left) and AFLUX (right). The fraction of convective (c,d) and large-scale (e,f) precipitation to the total precipitation is shown for co-located and separated components. The dashed vertical line at 70° N represents the minimum latitude that we use for all other analyses.

Throughout AFLUX, significant rainfall occurs mainly during the event of AR4 (Fig. 7). Also here, the rainfall is concentrated below 75° N (Fig. 8). Thus, for both campaigns, snow is the dominant type of precipitation north of 75° N (Fig. 8). The residual for the convective snow and rain as well as for the large-scale snowfall is about two times higher during ACLOUD

than during ThAFLUX (Table 3, Fig. 7).

The seasonal differences in the latitudinal distribution of precipitation between the early spring AFLUX and early summer ACLOUD campaigns are shown in Fig. 8. Rain plays a role for ACLOUD up to 75° N while hardly any rain occurs during AFLUX. For ACLOUD, ARs together with fronts bring snow into the Arctic up to 80° N while during AFLUX, cyclone-affected precipitation reaches up to the pole. The residual for the convective and large-scale snow is mainly found in the central

Arctic, a region where weather systems occur rarely. During AFLUX 100 % of the precipitation above 80° N is large-scale with the residual getting more important close to the pole. For the warmer ACLOUD campaign already up to 50 % of the residuals



close to the pole is convective. Nevertheless, the total precipitation amount is very low above 85° N and thus the question arises whether this is a real effect or due to model instabilities (see previous section).

### 3.6 Sensitivity of the results to the detection algorithms

All results discussed up to now have been achieved using the standard configuration with AR_Gu and CYC_S (GuS). But, how strongly do these results depend on the choice of algorithms? First, we investigate the difference between the two AR algorithms keeping CYC_S also in combination with AR_Go (GoS) (Fig. 9, top). During ACLOUD, we can see from Tab. 1 that much less precipitation is related to AR for GoS (19%; co-located: 16%; only: 3%) compared to GuS (40%; co-located: 31%; only: 9%). Thus, GuS mainly attributes precipitation frequently to fronts only (O-FRONTS). Furthermore, precipitation

only related to ARs (O-AR) in GuS is classified as residual by GoS, especially at the end of the campaign. However, while GuS led to a gap in the detection of AR1 on 30 May this is not the case for GoS which detects the AR continuously. This implies that IVT decreases within the AR and therefore the criteria in GuS to detect the AR are not fulfilled anymore but the IWV criterium still holds. During AFLUX, we see the opposite behaviour. GuS underestimates the precipitation related to ARs for all events during the campaign. Thus, GoS produces the strong precipitation contribution by cyclones discussed before, while

for GoS precipitation is most frequently related to AR-CYC. Consequently, the contribution of ARs would increase by 8% and the contribution of O-CYC would decrease by 6% (Table 1).

Regarding the latitudinal dependence (Fig. 10, top), we can see that the higher precipitation rates for GuS during ACLOUD, and GoS during AFLUX depend on the size of the area. Thus, during ACLOUD, the area of ARs detected by GuS is a factor of two higher than the area of ARs detected by GoS (Table 1). The largest deviation is between 70 and 75° N which

are the latitudes with the greatest precipitation rates (see Fig. 6). Consequently, the higher precipitation rates are associated with O-FRONTS or none of these systems (residual). During AFLUX, we can see the opposite effect. Applying the detection algorithm by GuS, ARs do not have a strong effect in higher latitudes. Consequently, the precipitation in the higher latitudes is mainly associated with O-CYC. However, when we apply the detection algorithm GoS, we can see that O-CYC is replaced by AR-CYC. Thus, the total contribution of ARs would increase from 16 (GuS) to 40% (GoS).

Second, we compare the two cyclone detection algorithms keeping AR_Gu for the AR detection (Fig. 9, top) (Fig. 9 and 10, bottom). The difference between GuS and GuA in terms of precipitation rate is not as strong as for the choice of the AR algorithm except AR4 where GuS has the AR connected with a cyclone while GuA attributes the precipitation to AR only. The reason lies in the much larger area which cyclones occupy in GuS compared to GuA during AFLUX. Here we can see a strong difference between both campaigns. During ACLOUD, the area of cyclones north of 70° N detected by GuS is smaller

compared to GuA whereas, during AFLUX, the area of cyclones is higher compared to GuA.

The area detected by the algorithm also influences the classification of the residual. This is best illustrated in the meridional distribution (Fig. 10). Up to 85° N a large part of the precipitation is assigned to be residual by GuA while it is contained in the cyclone category in GuS during AFLUX raising trust in the latter algorithm. However, we see the opposite albeit weaker behaviour during ACLOUD. Here, GuS produces some residual precipitation while this is assigned to cyclones in GuA.



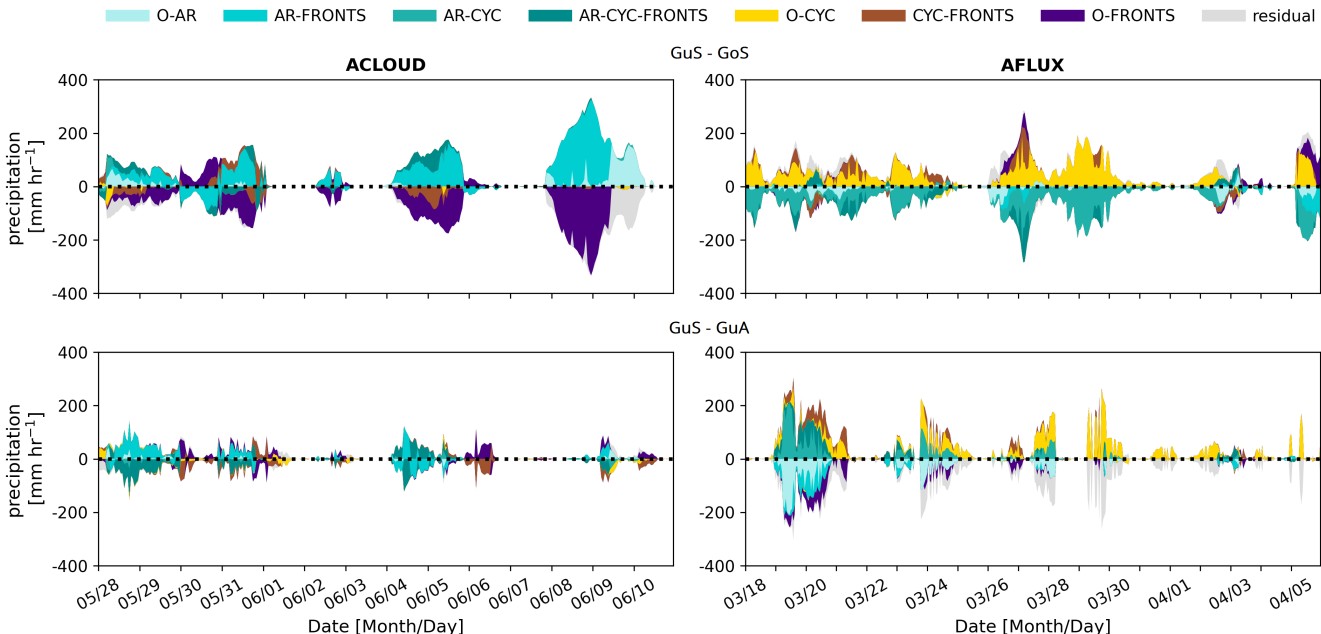

**Figure 9.** Time series for ACLOUD (left) and AFLUX (right) showing the difference of the contribution to precipitation by using different algorithms. Comparison of AR algorithms (GuS - GoS) ((a) and (b)), and comparison of cyclone algorithms (GuS - GuA) ((c) and (d)) for co-located and separated components (colors as in Fig. 1).

The choice of the algorithm has a strong effect on the assignment of different categories of precipitation. Table 1 illustrates how the distribution of precipitation to the different categories changes in terms of daily precipitation rate and area when different combinations of algorithms are considered. For all combinations, the area residual is larger than the precipitation residual. For ACLOUD our standard configuration (GuS) produces the lowest precipitation residual (28%) and is among the lower ones in terms of precipitation area. For AFLUX, GoS features the lowest residual for both precipitation and area. For this

campaign period, CYC_A produces a rather small cyclone area which especially in conjunction with AR_Gu (GuA) leads to a high residual of 38 % in precipitation rate and 57% in area. As already mentioned before, ACLOUD shows different behaviour. Here the residual precipitation area is especially high but does not vary too much between the different algorithms (62-68 %). This might indicate that weather systems are less important here and precipitation might also be produced locally as visible by the higher likelihood of convective precipitation for ACLOUD (Tab. 3).

**4   Conclusions**

We analysed the contribution of ARs, cyclones, and fronts to the total precipitation during two different periods, namely ACLOUD (early summer, May/June 2017) and AFLUX (early spring, March/April 2019). Both campaigns covered the northern North Atlantic sector which exhibits the highest precipitation rates in the Arctic. The two campaign periods differed from




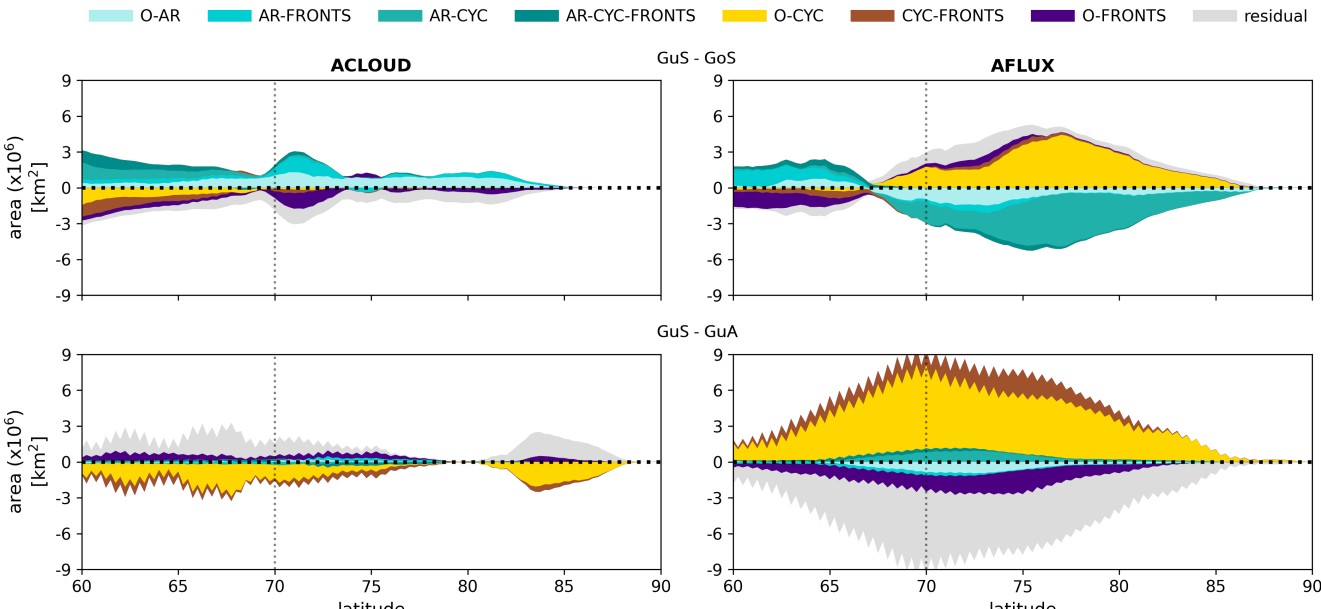

**Figure 10.** Latitudinal dependence for ACLOUD (left) and AFLUX (right) regarding the difference of the affected area by using different algorithms. Comparison of AR algorithms (GuS - GoS) ((a) and (b)), and comparison of cyclone algorithms (GuS - GuA) ((c) and (d)) for co-located and separated components (colors as in Fig. 1). The zigzag pattern in (c) and (d) is caused by the interpolation to the 0.25° ERA5 grid.

climatology in so far as localized hotspots of positive precipitation anomalies due to the weather systems and a drier central

Arctic occurred for ACLOUD, while AFLUX showed enhanced precipitation over most of the area.

We have established a new methodology that allows us to analyse the contribution of ARs, cyclones, and fronts to Arctic precipitation. As these features can be connected, we have defined seven different components: O-AR, AR-CYC, AR-FRONTS, AR-CYC-FRONTS, O-CYC, CYC-FRONTS, and O-FRONTS. Further, the precipitation rate which is not associated with any of these systems (so-called residual) is also taken into account. In its standard configuration the AR detection algorithm by

Guan et al. (2018) and the cyclone detection algorithm by Sprenger et al. (2017) is used. We tested the method over the two campaign periods in detail, having an application over the full ERA5 period in mind.

Although the campaign periods were chosen around the occurrence of ARs, we find that the precipitation related to ARs is not the main contributor to precipitation. During ACLOUD, precipitation is mainly associated with front-related components (55%) followed by AR-related components (40%), while cyclone-related components (22%) play a minor role. During

AFLUX, however, the precipitation is mainly associated with cyclone-related components (62%) and already 47 % of precipitation is only due to cyclones. AR and front-related components, 16%, and 19%, respectively play a minor role. While precipitation associated to cyclone related components is rather light during AFLUX, it shows a much higher daily averaged, area accumulated precipitation rate (12.5 x $10^3$ mm day$^{-1}$) compared to ACLOUD (7.6 x $10^3$ mm day$^{-1}$) due to their frequent





occurrence. Snow is the dominant form of precipitation being nearly exclusive for the colder AFLUX period (90%) than for
AFLUX (68%). Because ARs contribute more to rain than snowfall during both campaigns, any changes in AR characteristics
might be important for Arctic precipitation.

Several studies employ thresholds such as $0.1 \, \text{mm h}^{-1}$ (Boisvert et al., 2018) to eliminate "artificial" precipitation generated
by numerical models. Here, we did not use any threshold. However, we performed a sensitivity study in which we tested
different thresholds. In accordance with Boisvert et al. (2018), we stress the importance of trace precipitation (precipitation <
$0.1 \, \text{mm h}^{-1}$) for the Arctic; the introduction of a $0.1 \, \text{mm h}^{-1}$ threshold drastically reduces the total accumulated precipitation
by a factor of 2 (ACLOUD) and 3 (AFLUX). The higher the threshold the more light precipitation especially over the Arctic
Ocean disappears. Thus, the contribution of ARs connected with fronts and cyclones increases (by a factor of two), whereas
the residual decreases (by a factor of three) with higher thresholds, which might also hint at limits in the detection algorithms
as they are often not adapted to the Arctic.

We investigated the impact of the AR detection algorithm by comparing the standard setting (AR_Gu) with the AR_Go
algorithm by Gorodetskaya et al. (2014, 2020). For the latter, the precipitation contribution by ARs is reduced by a factor of
two for ACLOUD, compared to AR_Gu. The underestimation of ARs detected by AR_Go is mainly due to the shape, which
is a factor of two smaller, compared to the standard configuration. Thus, the precipitation is associated with one of the other
weather systems or it is classified as residual. For AFLUX the opposite effect occurs and more precipitation is attributed to ARs
when AR_Go is used. Comparing the contribution of cyclones when using the algorithm CYC_A by Akperov et al. (2007), we
can also see strong differences during the campaigns. During ACLOUD, cyclones detected by CYC_A cover a greater area,
which results in higher cyclone-associated precipitation compared to the standard configuration. The opposite effect occurs
during AFLUX: Here precipitation within cyclones detected by the standard configuration is higher compared to CYC_A.
These results highlight the importance of understanding the limitations of the underlying detection algorithms.

For the early spring period (AFLUX) we found much higher importance of cyclones for precipitation, while ARs dominate
in the early summer period (ACLOUD). However, for drawing robust conclusions about these seasonal differences, a long-
term assessment exploiting the full ERA5 record is planned in the future. Within this exercise, it might be possible to identify
changes in precipitation (phase) associated with different weather systems supporting a better understanding of the role of
airmass transport in the Arctic.

**A**

**A1**





**Figure A1.** Daily averaged precipitation rate [mm day$^{-1}$] for ACLOUD (left) and AFLUX (right). The dots represent for each pixel the contribution of ARs (turquoise) ((a) and (b)), cyclones (yellow) ((c) and (d)), and fronts (purple) ((e) and (f)) to the total precipitation. The grey dots indicate the residual fraction ((g) and (h)) which is not classified either as ARs, cyclones, or fronts. The increasing magnitude of the contribution (0-25%, 25-50%, 50-75%, and 75-100%) is shown by the increasing size of the dots.



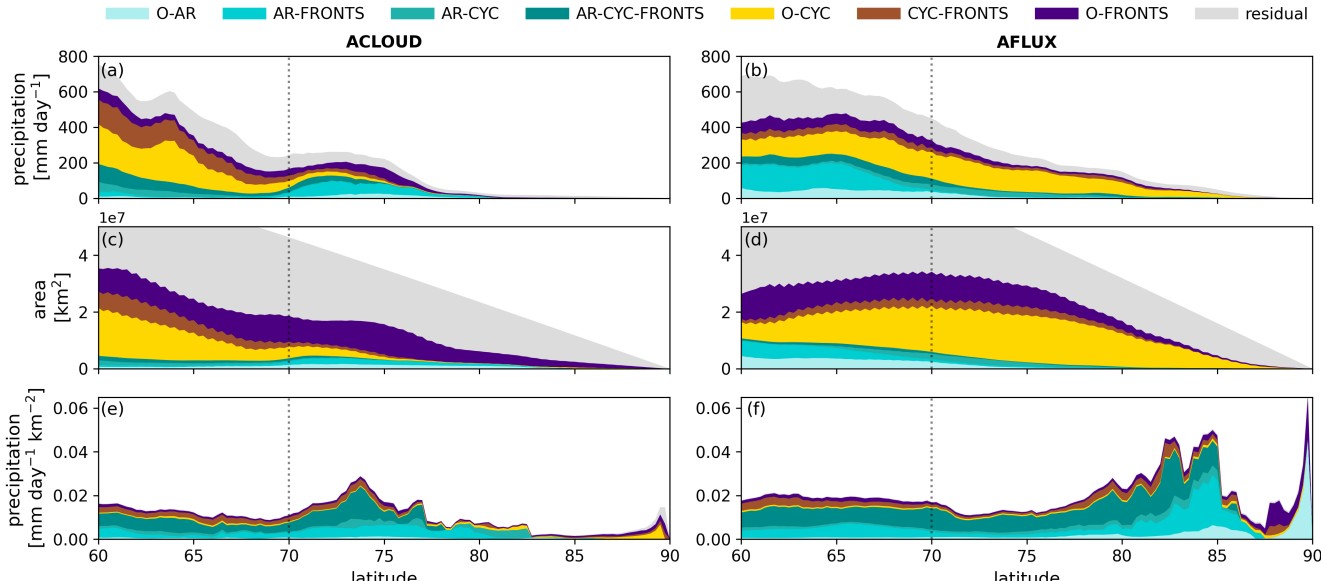

**Figure A2.** Latitudinal dependence (60 - 90° N) of daily averaged precipitation rate [mm day$^{-1}$] ((a) and (b)), the size of the area [km$^2$] ((c) and (d)), and the ratio between the precipitation rate and the area [mm day$^{-1}$ km$^{-2}$] ((e) and (f)) for ACLOUD (left) and AFLUX (right) for the co-located and separated components (colors as in Fig. 1). The dashed vertical line at 70° N represents the minimum latitude that we use for the other analyses.



**Figure A3.** Contribution of different precipitation rates [mm h$^{-1}$] to the daily averaged precipitation [mm day$^{-1}$] for co-located and separated components (a-h) during ACLOUD (top) and AFLUX (bottom). The accumulated daily precipitation rate [mm day$^{-1}$] is shown by the dotted line and their value is given by the y-axis. Note, that each 0.25 x 0.25 grid cell is area-weighted.



*Code availability.* The AR detection algorithms are provided by Bin Guan (see https://ucla.box.com/ARcatalog), and Irina Gorodetskaya (upon request).

*Data availability.* The reanalysis datasets used in this study were provided by ECMWF for ERA5 (https://doi.org/10.24381/cds.adbb2d47)
and (https://doi.org/10.24381/cds.bd0915c6); (Hersbach et al., 2018a, b)

.

*Video supplement.* The videos illustrate the temporal evolution of ARs during ACLOUD and AFLUX. The videos are available at the following link: https://uni-koeln.sciebo.de/s/mt0GuqpMPbKecT3

*Author contributions.* ML performed most analysis and produced all figures. ML and SC conceptualized the paper with the help of AR. IG
supported the AR analysis. MS provided the cyclone and frontal classification expertise. All authors contributed to the text.

*Competing interests.* The contact author has declared that neither they nor their co-authors have any competing interests

*Acknowledgements.* We gratefully acknowledge the funding by the Deutsche Forschungsgemeinschaft (DFG, German Research Foundation) – Project 268020496 – TRR 172, within the Transregional Collaborative Research Center "ArctiC Amplification: Climate Relevant Atmospheric and SurfaCe Processes, and Feedback Mechanisms (AC)3". AR was partly supported by the European Union's Horizon 2020
research and innovation framework program under Grant agreement no. 101003590 (PolarRES). We thank Mirseid Akperov for providing his cyclone classification with a 1-h resolution for the ACLOUD and AFLUX campaigns. This work used resources from the Deutsches Klimarechenzentrum (DKRZ) granted by its Scientific Steering Committee (WLA) under project ID bb1086. Further, we would like to thank Bin Guan for providing the second version of the AR detection algorithm.



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
