# Peer review of "Influence of atmospheric rivers and associated weather systems on precipitation in the Arctic"

_EGUsphere, 2023_

## Referee Comment (RC2)

Review of "Influence of atmospheric rivers and associated weather systems on precipitation in the Arctic" by Lauer et al.

**General Comments:**

The paper quantified the contributions of ARs, cyclones, and fronts to the total precipitation over the Arctic Atlantic sector during the two airborne campaigns, ACLOUD and AFLUX, using ERA5 data. It is found that during ACLOUD, AR- and front-related systems were most related to the precipitation rate. In contrast, during AFLUX, cyclone-related components played a dominant role in the precipitation rate. The authors further analyzed the precipitation types (convective or large-scale) and phases (snow or rain) associated with different systems. In addition, they quantified the uncertainty by comparing the results according to different AR and cyclone detection algorithms and the application of precipitation thresholds.

The paper was very interesting and mostly clear to read. The analysis was largely based on exploratory analysis. Linking ARs to Arctic precipitation and comparing their contributions with other weather systems using a new classification method seem appealing. It will be an important contribution to the AR literature. However, I have two main concerns which could be addressed and make the study clearer and more robust.

First, the authors consistently emphasized the assumption that seasonal differences account for the different results in the two campaigns. However, this work was mainly based on two short periods of airborne campaigns. It is too soon to attribute the different results during the two campaigns to seasonal differences. As the authors stated at the end of the paper, "for drawing robust conclusions about these seasonal differences, a long-term assessment exploiting the full ERA5 record is planned in the future". Without the long-term climatology study, I would not suggest consistently implying different results during the two campaigns owing to the seasonal differences. Specifically, in lines 269-270, the hypothesis was made without any explanation. Lines 276-277, 302-304, 366-367, and 455-456 consistently emphasized the "seasonal differences."

Second, the authors compared the results based on the different AR and cyclone detection algorithms in section 3.6. However, the authors did not explain why they should be so (such as Lines 445-454). Perhaps the authors can explain more about the observed different results based on the different algorithms, such as the different AR detection criteria between AR\_Gu and AR\_Go, the different physical aspects that each algorithm emphasized, and so on....

**Specific Comments:**

• Lines 49-51: I do not think that ARs "only cover about 10% of the Earth's surface circumference but are responsible for more than 90% of the poleward moisture transport in and across mid-latitudes" were found by Nash et al, (2018). In Nash et al, (2018), they cited that "Annually, ARs contribute over 90% of the poleward moisture

transport in the middle to high latitudes, despite only covering ~10% of the Earth's circumference over the midlatitudes (Guan & Waliser, 2015; Zhu & Newell, 1998)."

Zhu, Y., and R. E. Newell (1998), A proposed algorithm for moisture fluxes from atmospheric rivers, Mon. Weather Rev., 126, 725–735, doi:10.1175/1520-0493(1998)126<0725:APAFMF>2.0.CO;2.

Guan, B. and Waliser, D. E.: Detection of atmospheric rivers: Evaluation and application of an algorithm for global studies, JOURNAL OF GEOPHYSICAL RESEARCH-ATMOSPHERES, 120, 12 514–12 535, https://doi.org/10.1002/2015JD024257, 2015.

- Lines 121 and 123, 148,150: atmospheric data in ERA5 on the standard pressure levels from 1000 hPa to 1 hPa (i.e., a total of 37 vertical pressures) are interpolated from the 137 hybrid sigma/model levels in the Integrated Forecasting System (IFS). However, surface pressures over the Arctic study domain may be lower than 1000 hPa at high altitudes (e.g, Greenland). Therefore, considering Arctic topography, it is best that the integration is from the surface to 300 hPa. However, I think the current calculations of the integration from 1000 hPa would not change conclusions.
- Line 125: by "the 85th percentile of IVT", do you mean seasonally-based 85th percentile of IVT as stated in Guan and Waliser (2015)?
- Lines 197-198: "5-25 °E", for me, it seems around 15-25 °E?
- Lines 297-298: I do not quite understand the point in the "Therefore, it could be possible that parts of light precipitation related to residual (or also to the other weather systems) might be in the vicinity of the detected AR shape." Maybe you can explain more about that.
- In lines 302-304, can you expand on what "AR-related intensity" are being referred to here?
- Lines 338-339: based on Table 3, should the sentence be "clear dominance of cyclones for AFLUX (co-located: 14%, only: 48%) compared to ACLOUD (co-located: 12%, only: 7%)"?
- Line 347: I am unsure about the sentence, "For rain, the fraction of total precipitation is highest for ACLOUD with 33% and lower for AFLUX with 10%". I suggest rewriting.

- Line 379: I do not quite understand "Thus, GuS mainly attributes precipitation frequently to fronts only (O-FRONTS)." How do you conclude this based on Table 1?
- Line 384: "Thus, GoS produces the strong precipitation contribution by cyclones discussed before" do you refer to GuS, not "GoS" as in the text?
- Lines 385-386: Would you please expand on the expression "Consequently, the contribution of ARs would increase by 8%, and the contribution of O-CYC would decrease by 6%" to be more clear?
- Line 390: Do you mean Figure 8?
- Line 50: do you mean by many regions'?

---

## Author Comment (AC1)

**Answers to Reviewer 1**

We would like to thank the reviewer for his/her time and effort in reviewing our study. We have found the comments to be constructive and helpful and think that they have helped to make the aim of the study clearer! The comments of the reviewer are marked in black, and our answers to the reviewer are in blue. In the revised document, all new text is marked in blue, and deleted text is crossed out in red. Green refers to line numbers in the original manuscript.

1. Throughout the manuscript, the authors seem to implicitly assume that the difference between the two study periods are due to seasonality. For example, in lines 276, 302 and 366, the authors use the term "seasonal differences". However, the results presented here based only on two short periods: one is 14 days and another one is 19 days. Until further study based on the full ERA5 record is conducted, I would not recommend the authors attributing the differences between two study period to seasonal differences.

Thank you for this comment. We agree that the formulation is misleading and that we cannot conclude about general seasonality-driven differences based on the presented two cases. We have gone through the whole text and changed the associated text passages accordingly (L9, 12, 287, 294, 338, 398). Lines in the original text: 9, 12, 269, 276, 302, 366

2. Line 96: To me, the cyclone seems to locate in the Northeastern or Eastern Greenland.

Thanks for the hint. We change it to Eastern Greenland (L101) Line 96

3. Line 126-127: The authors should double-check the Guan & Waliser (2015) paper. I don't think they set the lower limit of the IVT threshold to 50 kg/m/s in the polar regions.

Yes, you are right. The value of 50 kg/m/s as a lower limit of the IVT threshold in the polar regions is set in the second version of their algorithm. We change this sentence to:

"However, due to the lower moisture capacity of the polar regions, the lower limit in these regions is set to > 50 kg/m/s in their second version (Guan et al. 2018)" (L134). L126-127

4. Line 184-186: "The effect is roughly 20 % of the residual and is slightly larger in absolute terms during 185 ACLOUD (about 8 % of the total precipitation) compared to AFLUX (about 5 %) as residual precipitation is more frequent during ACLOUD." This sentence needs to be rewritten for better clarity. I am not sure what the authors are trying to convey here.

The statement was rather brief and did not explain how the percentage numbers were derived. We reformulated and extended the text for clarification.

"Figure 2 illustrates how the residual precipitation declines if the frontal area, defined by the distance from the frontal zone, is reduced. Extending the distance from its shortest (139 km) to its largest (250 km) value, a drop in the residual of 8% for ACLOUD, and by 5% for AFLUX can be seen. This is basically independent of the precipitation threshold and the potential temperature gradient." (L197-200) L184-186.

5. Line 269-274. More explanations for the hypothesis are needed. The two points followed the first sentence in this paragraph are simply describing the difference between the two periods. They are not explanations for the hypothesis.

We agree and have reformulated this according sentence to:

"In summary, we can highlight two differences between the campaign periods." (L287) L269-274

6. Line 289: How do you weigh precipitation rates by the area? To me, you just simply sum up the precipitation rates across all the grid points over the study region.

Yes, for the histogram analysis (L307-323), we do weigh precipitation rates by the area. However, for all other analysis we do so. To make this more clear, we added a sentence in the method section L111-112:

"Because the area of an ERA5 grid cell decreases towards the North Pole, we take this effect into account when precipitation over larger areas is considered. Therefore, if noted otherwise, the area-wide precipitation averages are computed as an area-weighted average."

7. Table 1 caption: For the daily averaged precipitation rate, did you calculate it simply by summing up the precipitation rates across all grid points over the study region?

First, we weight the precipitation by the area (see above). Then we sum up the precipitation rates across all grid cells over the study region and for the whole period. We added 'area- weighted' in the caption to make clear that the precipitation rates are weighted beforehand.

8. Line 304: In my opinion, meridionally oriented ARs over the Arctic should be more effective in inducing precipitation. Meridionally oriented ARs over the Arctic travel across a strong temperature gradient from warmer regions to colder regions. Cold air holds less moisture. The moisture inside the ARs thus have to precipitate out. That is to say, for a given AR and a given season, the AR would induce stronger precipitation when it orients meridionally compared to when it orients zonally.

Thank you for this comment. We had a detailed look into it and modified the paragraph accordingly, also briefly discussing the aspect mentioned above. (L330 - 338)

"Comparing both campaigns, there are differences regarding AR-related precipitation intensity. During ACLOUD, AR events caused several maxima in precipitation rate while for AFLUX only AR4 brings significant precipitation into the studied domain (Fig. 4). AR4 was a meridionally orientated AR and reached up to 87° N. It crossed a strong temperature gradient when crossing the sea ice edge around 77° N. Thus, it experienced a strong drop in moisture saturation, which led to the release of precipitation. While other ARs during AFLUX were also meridionally orientated, they did not reach that far north or touched the studied domain only marginally. Therefore, their contribution to the total precipitation in the studied region is comparable low. The higher precipitation amount during ACLOUD is mainly due to the higher number of ARs at higher latitudes – two of them (AR1 and AR2) were zonally orientated (see videos in the supplement (Lauer, 2022)). Together, ACLOUD and AFLUX provide a variety of AR appearances to test our methodology, but long-term studies are needed to detect seasonal differences."

9. Line 347: "For rain, the fraction of total precipitation is highest for ACLOUD with 33% and lower for AFLUX with 10%." This sentence needs to be rewritten for better clarity.

**We have rewritten the sentence as:**

"The fraction of rain to the total precipitation is higher for ACLOUD (33%; convective: 10%, large-scale: 23%) than for AFLUX (10%; convective: 5%, large-scale: 5%)." (L379-380)

10. Line 367: Based on Figure 8b, rain occurs during AFLUX.

You are right, we improved the discussion:

"The differences in the latitudinal distribution of precipitation between the early spring AFLUX and early summer ACLOUD campaign periods are shown in Fig. 8. While rain occurs up to 75°N for both periods, significant amounts of snow reach higher latitudes during AFLUX (up to 85°N) compared to ACLOUD (up to 78°N)." (L398-401)

11. Line 383: I would not use "underestimate" here. We don't know which AR detection algorithms represent the truth.

We agree, and have reformulated the according sentence by using "... is lower ...".

The precipitation related to ARs is lower for GuS than for GoS for all events during this campaign. (L421-422)

12. Line 385-386: I couldn't find these 8% and 6% in Table 1.

Thank you for this comment. Unfortunately, we extracted the values from the wrong line. We corrected the sentence as follows:

"Thus, the contribution of the AR- and cyclone-related components differ among the algorithms. Consequently, for GoS the contribution of ARs is 24% (co-located: 20%, only 4%) higher, whereas the contribution of O-CYC is 18% lower compared to GuS." (L425-427)

**13.** Line 390: "Consequently, ..." I don't see how previous sentences can lead to this conclusion. More explanations are needed.

We changed the beginning of this sentence to "In this area,..":

"The largest deviation is between 70 and 75° N which are the latitudes with the greatest precipitation rates (see Fig. 8). In this area, the higher precipitation rates are associated with O-FRONTS or none of these systems (residual). " (L430-432)

14. Line 445-454: Any potential explanations for why AR\_GU detects larger ARs for ACLOUD, but smaller ARs for AFLUX, and why CYC\_A detects larger cyclones during ACLOU, but smaller cyclones during AFLUX?

Thank you for this comment. An extensive algorithm intercomparison is beyond the scope of the manuscript. However, we added some explanations for both AR and cyclone effects.

For Atmospheric Rivers:

We produced a new Figure A4 which shows the time series of domain-accumulated hourly precipitation as in Fig.4, but for GoS:

Figure A4. Time series of domain-accumulated hourly precipitation rate [mm h-1] (a,b), the size of the area [km2] (c,d), and the ratio between the precipitation rate and the area [mm h-1 km-2] (e,f) for different weather systems for ACLOUD (left, 28 May - 11 June 2017)

and AFLUX (right, 18 March - 6 April 2019) for GoS. The colors represent the co-located and separated components.

We added new text sections to clarify the differences:

- In Fig. 9, we can see that the precipitation which is related to AR-FRONTS for GuS is mainly attributed to fronts only (O-FRONTS) for GoS. This is possible because the threshold in AR\_Go is based on the IWV, thus only on the moisture content that is reduced by precipitation. Since the AR is typically found in the pre-cold frontal zone, the precipitation associated with the AR is defined as frontal precipitation at the time when the AR is longer defined. (L413-417)
- In general, the greatest amount of precipitation during AFLUX is classified as light precipitation (Fig. 5). Therefore, we assume that the moisture content is too low and the threshold of AR\_Gu cannot be exceeded in the higher latitudes (Fig. 10). (L433-435)
- We investigated the impact of the AR detection algorithm by comparing the standard setting (AR Gu) with the AR Go algorithm by Gorodetskaya et al. (2014, 2020). Comparing both algorithms, we can highlight two differences. First, AR Gu uses IVT (humidity and wind), whereas AR\_Go uses IWV and IWVsat (humidity and temperature). Second, although both algorithms make use of a threshold, these thresholds differ conceptually. Due to the different concepts of the algorithms, we can see differences in the time period, the area, and the precipitation amount associated with ARs (Tab. 1, Figs. 4, 9, 10, and A4). During ACLOUD, the area, as well as the amount of AR-related precipitation, is a factor of two higher for AR Gu compared to AR Go (Tab. 1). Especially precipitation rates associated with ARs and fronts are affected (Figs. 9 and 10), e.g. for AR2, AR\_Go detects a more confined AR area, while AR Gu broadened this AR area by the comma head of the cyclone and the frontal precipitation. For AFLUX, the opposite effect occurs. During this campaign period, the precipitation rate, as well as the area is more than a factor of two higher for AR\_Go than AR\_Gu. Here, especially precipitation rates associated with ARs and cyclones are affected (Figs. 9 and 10). Here, we assume that the moisture content is too low and the threshold of AR\_Gu cannot be exceeded in the higher latitudes (Fig. 10), while AR\_Go is specifically tailored to the relatively dry conditions of the high latitudes. In summary, based

on the limited campaign periods, we cannot conclude about the generality of the differences. Therefore, a long-term statistical analysis is needed. (L494-507)

For cyclones:

"These differences could be the consequence of different pressure intervals to detect the outermost closed isobar and elevation filters (described in section 2.2.2). Generally, the higher (coarser) pressure interval for CYC\_S (0.5 hPa) could reduce the size of the cyclone, compared to CYC\_A which uses a smaller pressure interval of 0.1 hPa. This explains, that CYC\_A detects larger cyclones and cyclone-associated precipitation during ACLOUD. In addition, different elevation filters in CYC\_S and CYC\_A affect cyclone detection and related precipitation." (L511-515)

15. Line 50: regions' hydroclimate? Yes, we corrected this typo. (L52)

16. Line 384: Do you mean GuS produced the strong precipitation contribution by cyclones? Yes. We changed it to GuS. (L422)

17. Line 435: Do you mean ACLOUD? Yes, we changed it to ACLOUD (L478)

---

## Author Comment (AC2)

**Answers to Reviewer 2**

We would like to thank the reviewer for his time and effort in reviewing our study. We have found the comments to be constructive and helpful and think that they have helped to make the aim of the study more clear! The comments of the reviewer are marked in black, and our answers to the reviewer are in blue. In the revised document, all new text is marked in blue, and deleted text is crossed out in red. Green refers to line numbers in the original manuscript.

**General comments**

First, the authors consistently emphasized the assumption that seasonal differences account for the different results in the two campaigns. However, this work was mainly based on two short periods of airborne campaigns. It is too soon to attribute the different results during the two campaigns to seasonal differences. As the authors stated at the end of the paper, "for drawing robust conclusions about these seasonal differences, a long-term assessment exploiting the full ERA5 record is planned in the future". Without the long-term climatology study, I would not suggest consistently implying different results during the two campaigns owing to the seasonal differences. Specifically, in lines 269-270, the hypothesis was made without any explanation. Lines 276-277, 302-304, 366-367, and 455-456 consistently emphasized the "seasonal differences."

Thank you for this comment. We agree that the formulation is misleading and that we cannot conclude about general seasonality-driven differences based on the presented two cases. We have gone through the whole text and changed the associated text passages accordingly (L9, 12, 287, 294, 338, 398). Lines in the original text: 9, 12, 269, 276, 302, 366

Second, the authors compared the results based on the different AR and cyclone detection algorithms in section 3.6. However, the authors did not explain why they should be so (such as Lines 445-454). Perhaps the authors can explain more about the observed different results based on the different algorithms, such as the different AR detection criteria between AR\_Gu and AR\_Go, the different physical aspects that each algorithm emphasized, and so on....

Thank you for this comment. An extensive algorithm intercomparison is beyond the scope of the manuscript. However, we added some explanations for both AR and cyclone effects.

For Atmospheric Rivers:

We produced a new Figure A4 which shows the time series of domain-accumulated hourly precipitation as in Fig.4 , but for GoS:

Figure A4. Time series of domain-accumulated hourly precipitation rate [mm h-1] (a,b), the size of the area [km2] (c,d), and the ratio between the precipitation rate and the area [mm h-1 km-2] (e,f) for different weather systems for ACLOUD (left, 28 May - 11 June 2017) and AFLUX (right, 18 March - 6 April 2019) for GoS. The colors represent the co-located and separated components.

We added new text sections to clarify the differences:

- In Fig. 9, we can see that the precipitation which is related to AR-FRONTS for GuS is mainly attributed to fronts only (O-FRONTS) for GoS. This is possible because the threshold in AR\_Go is based on the IWV, thus only on the moisture content that is reduced by precipitation. Since the AR is typically found in the pre-cold frontal zone, the precipitation associated with the AR is defined as frontal precipitation at the time when the AR is longer defined. (L413-417)
- In general, the greatest amount of precipitation during AFLUX is classified as light precipitation (Fig. 5). Therefore, we assume that the moisture content is too low and the threshold of AR\_Gu cannot be exceeded in the higher latitudes (Fig. 10). (L433-435)
- We investigated the impact of the AR detection algorithm by comparing the standard setting (AR\_Gu) with the AR\_Go algorithm by Gorodetskaya et al. (2014, 2020). Comparing both algorithms, we can highlight two differences. First, AR\_Gu uses IVT (humidity and wind), whereas AR\_Go uses IWV and IWVsat (humidity and temperature). Second, although both algorithms make use of a threshold, these thresholds differ conceptually. Due to the different concepts of the algorithms, we can see differences in the time period, the area, and the precipitation amount associated with ARs (Tab. 1, Figs. 4, 9, 10, and A4).
  During ACLOUD, the area, as well as the amount of AR-related precipitation, is a factor of two higher for AR\_Gu compared to AR\_Go (Tab. 1). Especially precipitation rates associated with ARs and fronts are affected (Figs. 9 and 10), e.g. for AR2,AR\_Go detects a more confined AR area, while AR\_Gu broadened this AR area by the comma head of the cyclone and the frontal precipitation. For AFLUX, the opposite effect occurs. During this campaign period, the precipitation rate, as well as the area is more than a factor of two higher for AR Go than AR Gu. Here, especially

precipitation rates associated with ARs and cyclones are affected (Figs. 9 and 10). Here, we assume that the moisture content is too low and the threshold of AR\_Gu cannot be exceeded in the higher latitudes (Fig. 10), while AR\_Go is specifically tailored to the relatively dry conditions of the high latitudes. In summary, based on the limited campaign periods, we cannot conclude about the generality of the differences. Therefore, a long-term statistical analysis is needed. (L494-507)

**For cyclones:**

"These differences could be the consequence of different pressure intervals to detect the outermost closed isobar and elevation filters (described in section 2.2.2). Generally, the higher (coarser) pressure interval for CYC\_S (0.5 hPa) could reduce the size of the cyclone, compared to CYC\_A which uses a smaller pressure interval of 0.1 hPa. This explains, that CYC\_A detects larger cyclones and cyclone-associated precipitation during ACLOUD. In addition, different elevation filters in CYC\_S and CYC\_A affect cyclone detection and related precipitation." (L511-515)

**Specific Comments:**

• Lines 49-51: I do not think that ARs "only cover about 10% of the Earth's surface circumference but are responsible for more than 90% of the poleward moisture transport in and across mid-latitudes" were found by Nash et al, (2018). In Nash et al, (2018), they cited that "Annually, ARs contribute over 90% of the poleward moisture transport in the middle to high latitudes, despite only covering ~10% of the Earth's circumference over the midlatitudes (Guan & Waliser, 2015; Zhu & Newell, 1998)."

Zhu, Y., and R. E. Newell (1998), A proposed algorithm for moisture fluxes from atmospheric rivers, Mon. Weather Rev., 126, 725–735, doi:10.1175/1520-0493(1998)1262.0.CO;2.

Guan, B. and Waliser, D. E.: Detection of atmospheric rivers: Evaluation and application of an algorithm for global studies, JOURNAL OF GEOPHYSICAL RESEARCH-ATMOSPHERES, 120, 12 514–12 535, https://doi.org/10.1002/2015JD024257, 2015.

Yes, you are right. We changed this sentence and added the correct references.

"Although ARs only cover about 10% of the Earth's surface circumference at midlatitudes, they are responsible for more than 90% of the poleward moisture transport in and across these latitudes." (L50-51)

• Lines 121 and 123, 148,150: atmospheric data in ERA5 on the standard pressure levels from 1000 hPa to 1 hPa (i.e., a total of 37 vertical pressures) are interpolated from the 137 hybrid sigma/model levels in the Integrated Forecasting System (IFS). However, surface pressures over the Arctic study domain may be lower than 1000 hPa at high altitudes (e.g, Greenland). Therefore, considering Arctic topography, it is best that the integration is from the surface to 300 hPa. However, I think the current calculations of the integration from 1000 hPa would not change conclusions.

Yes, we consider the Arctic topography by using the nearest surface level. To make it more clear, we modified Section 2.1:

"Specific humidity from 1000 hPa (or the nearest surface level) to 300 hPa is used to calculate the integrated water vapor (IWV), and together with horizontal wind components the integrated water vapor transport (IVT)." (L105 – 108)

Further, we add some text in Section 2.2.1

L127: (from  $p_1$  (1000 hPa, or the nearest surface level) to 300 hPa) L128, L130, L158, L160: We changed the boundaries of the integral to  $p_1$  and  $p_2$

• Line 125: by "the 85th percentile of IVT", do you mean seasonally-based 85th percentile of IVT as stated in Guan and Waliser (2015)?

Yes, you are right. We forgot to mention the percentile is calculated for each month. We added this in line 125:

"In the first version of their algorithm, Guan and Waliser (2015) first calculate the monthly-based 85th percentile of IVT for each grid cell from 1997 - 2014." (L131-133)

• Lines 197-198: "5-25 °E", for me, it seems around 15-25 °E?

Thanks, this is a typo. We have changed it to 15-25°E. (L211)

• Lines 297-298: I do not quite understand the point in the "Therefore, it could be possible that parts of light precipitation related to residual (or also to the other weather systems) might be in the vicinity of the detected AR shape." Maybe you can explain more about that.

At this point, we seek an explanation of why light precipitation is rare for grid cells classified as AR. Maybe the reference to other weather systems is confusing at this point. Therefore, we simplified the statement to:

"The reason for the rare occurrence of light precipitation might be the strict AR detection focusing on the innermost AR area. Precipitation that is still connected to the AR but occurring outside the AR shape is likely lower than precipitation in the core area. This would be a similar effect as for fronts where a reduced frontal area leads to an increase in the residual (Fig. 2)." (L318-321)

• In lines 302-304, can you expand on what "AR-related intensity" are being referred to here?

We try to make it more clear by referring to Fig. 4. (L332)

• Lines 338-339: based on Table 3, should the sentence be "clear dominance of cyclones for AFLUX (co-located: 14%, only: 48%) compared to ACLOUD (co-located: 12%, only: 7%)"?

Yes, you are right. We improved this. (L369-370)

• Line 347: I am unsure about the sentence, "For rain, the fraction of total precipitation is highest for ACLOUD with 33% and lower for AFLUX with 10%". I suggest rewriting.

We have rewritten the sentence:

"The fraction of rain to the total precipitation is higher for ACLOUD (33%; convective: 10%, large-scale: 23%) than for AFLUX (10%; convective: 5%, large-scale: 5%)." (L379-380)

• Line 379: I do not quite understand "Thus, GuS mainly attributes precipitation frequently to fronts only (O-FRONTS)." How do you conclude this based on Table 1?

Unfortunately, we mentioned Table 1 but it should be Figure 9. We have rewritten the sentence to make it more clear:

"In Fig. 9, we can see that the precipitation which is related to AR-FRONTS for GuS is mainly attributed to fronts only (O-FRONTS) for GoS. This is possible because the threshold in AR\\_Go is based on the IWV, thus only on the moisture content that is reduced by precipitation. Since the AR is typically found in the pre-cold frontal zone, the precipitation associated with the AR is defined as frontal precipitation at the time when the AR is no longer defined. " (L413-417)

• Line 384: "Thus, GoS produces the strong precipitation contribution by cyclones discussed before" do you refer to GuS, not "GoS" as in the text?

Yes. We changed it to GuS (L422)

• Lines 385-386: Would you please expand on the expression "Consequently, the contribution of ARs would increase by 8%, and the contribution of O-CYC would decrease by 6%" to be more clear?

Thank you for this comment. Unfortunately, we extracted the values wrong line. We corrected the sentence:

"Thus, the contribution of the AR- and cyclone-related components differ among the algorithms. Consequently, for GoS the contribution of ARs is 24% (co-located: 20%, only 4%) higher, whereas the contribution of O-CYC is 18% lower compared to GuS."(L425-427)

• Line 390: Do you mean Figure 8?

Thank you for catching this. Yes, it should be Figure 8. (L431)

• Line 50: do you mean by many regions'?

Yes, we correct this typo. (L52)

---

## Author Response (AR2)

**Answer to Hailong Wang**

**Comment:**

Thank you for submitting the revised manuscript. While all the comments and concerns from both referees are addressed, I found the use of units (mm/day or mm/h) for total precipitation in the study domain is confusing, for example, in Figures 4-8, Table 1 & 3 and the main text. It appears to be a sum of the precipitation rate in all grids within the domain, which I don't believe is often used for total regional precipitation. It might need to be converted to total mass of water with the area information included in the calculation. Please revised the figures, tables, and text accordingly.

**Answer:**

Dear Hailong Wang,

we greatly appreciate your comment. As far as we know, the specification in mm/h or mm/day over a larger area is a common method. We specifically screened the literature relevant for our topic and found that Boisvert et al., 2018; Gimeno et al. 2015; McCrystall et al., 2021; Rüdisühli et al., 2018; Viceto et al., 2022, (all cited in our manuscript), Pithan and Jung (2021), Boisvert et al. (2021), as well as the Atmospheric river studies by Gimeno-Sotelo et al. (2018, 2019) use the same units for precipitation as in our study. The only exceptions were the studies carried out by Richard Bintanja where the amount of precipitation is multiplied by the area to get the unit $km^3$. Since most publications use the unit mm/h or mm/day and also none of the reviewers addressed this point we would prefer to keep the unit in our study as it is.

Sincerely
Melanie Lauer – in the name of all authors

New references:

Boisvert, L., Grecu, M., & Shie, C. L. (2021). Investigating Wintertime GPM-IMERG Precipitation in the North Atlantic. *Geophysical Research Letters*, *48*(20), e2021GL095391. DOI: 10.1029/2021GL095391

Gimeno-Sotelo, L., Nieto, R., Vázquez, M., & Gimeno, L. (2018). A new pattern of the moisture transport for precipitation related to the drastic decline in Arctic sea ice extent. *Earth System Dynamics*, *9*(2), 611-625. DOI: 10.5194/esd-9-611-2018

Gimeno-Sotelo, L., Nieto, R., Vázquez, M., & Gimeno, L. (2019). The role of moisture transport for precipitation in the inter-annual and inter-daily fluctuations of the Arctic sea ice extension. *Earth System Dynamics*, *10*(1), 121-133. DOI: 10.5194/esd-10-121-2019

Pithan, F., & Jung, T. (2021). Arctic amplification of precipitation changes – The energy hypothesis. *Geophysical Research Letters*, *48*(21), e2021GL094977. DOI:10.1029/2021GL094977